# POIL: Preference Optimization for Imitation Learning

## Abstract

Imitation learning (IL) enables agents to learn policies by mimicking expert demonstrations. While online IL methods require interaction with the environment, which is costly, risky, or impractical, offline IL allows agents to learn solely from expert datasets without any interaction with the environment. In this paper, we propose Preference Optimization for Imitation Learning (POIL), a novel approach inspired by preference optimization techniques in large language model alignment. POIL eliminates the need for adversarial training and reference models by directly comparing the agent's actions to expert actions using a preference-based loss function. We evaluate POIL on MuJoCo control tasks under two challenging settings: learning from a single expert demonstration and training with different dataset sizes (100%, 10%, 5%, and 2%) from the D4RL benchmark. Our experiments show that POIL consistently delivers superior or competitive performance against state-of-the-art methods in the past, including Behavioral Cloning (BC), IQ-Learn, DMIL, and O-DICE, especially in data-scarce scenarios, such as using one expert trajectory or as little as 2% of the full expert dataset. These results demonstrate that POIL enhances data efficiency and stability in offline imitation learning, making it a promising solution for applications where environment interaction is infeasible and expert data is limited.

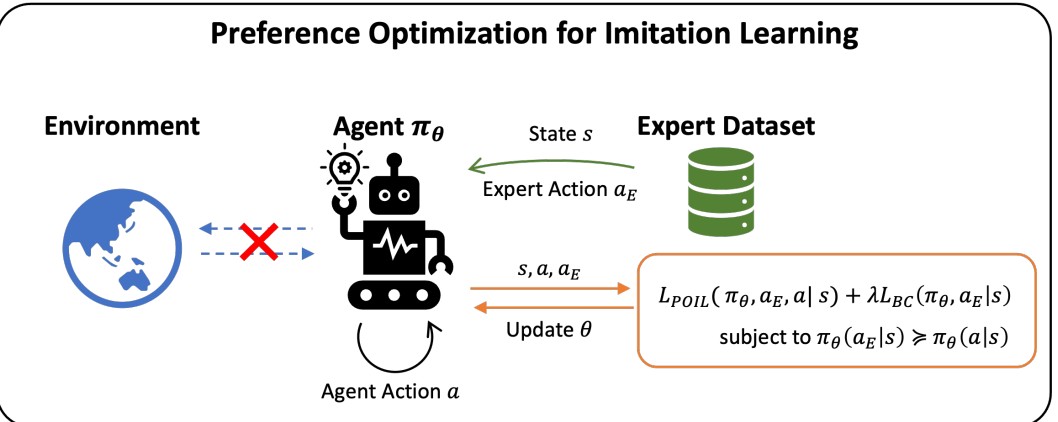

Figure 1: Process overview of preference optimization for imitation learning (POIL). The agent, guided by an expert dataset, compares the agent's actions with expert actions and computes the preference loss for updating the policy parameters. This process does not require environmental interaction, as the red cross indicates, and will be discussed in more detail in Subsection 3.3

# 1 INTRODUCTION

*Reinforcement learning* (RL) (Sutton, 2018) has achieved remarkable success across various domains, including video games (Mnih et al., 2015; Schrittwieser et al., 2020), robotics (Kober et al., 2013), and even nuclear fusion control (Degrave et al., 2022). However, defining suitable reward functions remains a significant challenge (Eschmann, 2021), especially in tasks where desired behaviors are abstract or hard to specify, such as control problems (Kiumarsi et al., 2017). Poorly designed reward functions can lead to unintended or unsafe behaviors (Amodei et al., 2016), and deep RL algorithms are often sensitive to reward sparsity (Ladosz et al., 2022), complicating the development of effective reward signals.

*Imitation learning* (IL) (Zare et al., 2024; Hussein et al., 2017) offers an alternative by learning policies directly from expert demonstrations without requiring explicit reward functions. In *online IL*, the agent interacts with the environment to learn the expert's behavior. Prominent methods like generative adversarial imitation learning (GAIL) (Ho & Ermon, 2016), adversarial inverse reinforcement learning (AIRL) (Fu et al., 2017), and discriminator actor-critic (DAC) (Kostrikov et al., 2018) employ a generator (the policy) and a discriminator (distinguishing between expert and agent behaviors) in an adversarial setup to encourage the agent to mimic the expert closely (Garg et al., 2021). Despite their effectiveness, these methods face practical challenges: the adversarial optimization process can be unstable and difficult to train, leading to biased, high-variance gradient estimators and convergence issues (Garg et al., 2021). Moreover, the need for environment interaction makes them impractical in real-world scenarios where such interaction is costly, risky, or infeasible (Lyu, 2024; Prudencio et al., 2023).

To address these limitations, *offline IL* methods have been developed to learn from pre-collected expert demonstrations without environment interaction. Behavior cloning (BC) (Pomerleau, 1991) is a straightforward approach that directly replicates expert actions through supervised learning. However, BC suffers from compounding errors due to distribution shifts and often requires large amounts of expert data (Ross et al., 2011). Recent advances aim to mitigate these issues by correcting for distribution discrepancies. The DICE family of algorithms, including ValueDICE (Kostrikov et al., 2019), DemoDICE (Kim et al., 2022), and O-DICE (Mao et al., 2024), improve upon BC by addressing distribution shift.

In this paper, we propose a novel offline imitation learning method called *preference optimization for imitation learning* (POIL), inspired by recent advances in preference-based alignment techniques in large language models (LLMs) (Wang et al., 2024), and clearly different from previous offline IL methods. An overview of the POIL process is illustrated in Figure 1, described in more detail in Subsection 3.3. Specifically, we draw inspiration from direct preference optimization (DPO) (Rafailov et al., 2024), contrastive preference optimization (CPO) (Xu et al., 2024), and self-play fine-tuning (SPIN) (Chen et al., 2024). These methods have been successful in aligning LLMs with human preferences but have specific requirements that limit their direct application to offline IL, namely, DPO and CPO require preference datasets and, in the case of DPO, a reference model. SPIN leverages an expert dataset but still relies on a reference model during training.

In contrast, POIL adapts these techniques to the offline IL setting by introducing a framework that directly compares the agent's actions to expert actions without the need for a discriminator or a reference model, as illustrated in Figure 1. By eliminating the need for preference datasets and reference models, POIL simplifies the learning process, avoids adversarial training instabilities, and enhances computational efficiency. This approach allows POIL to effectively overcome key constraints of existing methods in offline IL while improving overall performance.

We evaluate POIL on standard MuJoCo (Todorov et al., 2012) control tasks, including *HalfCheetah*, *Hopper*, and *Walker2d*. POIL consistently delivers superior or competitive results compared to state-of-the-art methods across various dataset sizes, particularly excelling in data-scarce scenarios, e.g., a single demonstration or small fractions of the dataset.

Our contributions are summarized as follows:

- We introduce POIL, a novel offline imitation learning method that eliminates the need for adversarial training, preference datasets, and reference models by directly comparing agent and expert actions.

- We provide empirical evidences on MuJoCo tasks showing that POIL achieves superior or competitive performance against state-of-the-art methods in the past, especially in data-limited scenarios.
- We conduct ablation studies to analyze the impact of some key hyper-parameters on POIL's performance, providing insights into its robustness and applicability.

These results suggest that preference optimization techniques from LLM alignment are effectively adapted to offline imitation learning, opening new avenues for research and applications in control and robotics.

## 2 RELATED WORK

### 2.1 OFFLINE IMITATION LEARNING

Offline imitation learning methods (Zare et al., 2024; Hussein et al., 2017) learn from static datasets without needing interaction with the environment. An early approach, behavior cloning (Pomerleau, 1991), learns directly from expert demonstrations but has issues with compounding errors and distribution shift, especially with limited data (Ross et al., 2011).

To overcome these issues, the DICE (DIstribution Correction Estimation) family of algorithms provides improvements. ValueDICE (Kostrikov et al., 2019) minimizes the KL divergence between stationary distributions, while SoftDICE (Sun et al., 2021) employs the Earth-Mover distance for distribution matching. DemoDICE (Kim et al., 2022) can use demonstrations of varying quality, and ODICE (Mao et al., 2024) adds orthogonal-gradient updates to handle conflicting gradients in learning. Other methods include adaptations of *inverse reinforcement learning* for offline use (Zolna et al., 2020; Yue et al., 2023), energy-based models (Jarrett et al., 2020), and OTR (Luo et al., 2023). Despite these advances, challenges remain. Some methods, such as DemoDICE and SMODICE (Ma et al., 2022), need extra data beyond expert demonstrations. Others struggle with limited datasets or single demonstrations.

### 2.2 PREFERENCE-BASED REINFORCEMENT LEARNING

Preference-based RL (PbRL) (Wirth et al., 2017) has emerged as a promising approach to address the challenges of reward function design in traditional RL by incorporating human preferences into the learning process. Early work by Christiano et al. (2017) demonstrated the potential of PbRL using deep learning techniques. This pioneering work opened new avenues for tackling challenging domains but relied heavily on external preference feedback. Subsequent research focused on reducing this dependency. Lee et al. (2021) introduced PEBBLE, which improved feedback efficiency through experience relabeling and unsupervised pre-training. Park et al. (2022) further improved this with SURF, a semi-supervised approach leveraging data augmentation, yet neither fully eliminated the need for human feedback.

A significant advancement in the field came with the inverse preference learning (IPL) proposed by Hejna & Sadigh (2024). IPL represents a novel approach specifically designed for learning from offline preference data. It leverages the key insight that for a fixed policy, the Q-function encodes all necessary information about the reward function. Using the Bellman operator, IPL eliminates the need for an explicit reward model, thereby simplifying the algorithm and improving parameter efficiency. This innovation marks a crucial step towards more efficient and scalable PbRL methods. However, IPL still requires an external preference dataset and remains a value-based method, leaving room for further improvements in data efficiency and algorithmic approach.

### 2.3 ALIGNMENT TECHNIQUES IN LARGE LANGUAGE MODELS

Reinforcement learning from human feedback (RLHF) (Kaufmann et al., 2023; Wang et al., 2024) has emerged as a powerful approach for aligning large language models (LLMs) with human preferences. Pioneered by InstructGPT (Ouyang et al., 2022) and further developed by Bai et al. (2022), RLHF involves training a reward model based on human preference data and then optimizing the policy using reinforcement learning guided by this reward model. While effective, RLHF is computationally intensive and requires careful hyper-parameter tuning.

To address these challenges, a class of methods known as DPO-like methods has been proposed as simpler alternatives that bypass explicit reward modeling. DPO (Rafailov et al., 2024) directly optimized the policy to match preference data using a classification loss. The standard DPO loss function is defined as:

$$\mathcal{L}_{\text{DPO}}(\pi_\theta; \pi_{\text{ref}}) = -\mathbb{E}_{(x,y_w,y_l)\sim\mathcal{D}} \left[ \log \sigma \left( \beta \left( \log \frac{\pi_\theta(y_w|x)}{\pi_{\text{ref}}(y_w|x)} - \log \frac{\pi_\theta(y_l|x)}{\pi_{\text{ref}}(y_l|x)} \right) \right) \right], \quad (1)$$

where $\pi_\theta$ is the agent's policy parameterized by $\theta$, $\pi_{\text{ref}}$ is the reference model, $\sigma(z) = 1/(1 + e^{-z})$ is the sigmoid function, $\beta$ is a scaling factor, $\mathcal{D}$ is the dataset of preference pairs $(x, y_w, y_l)$, and $y_w$ and $y_l$ denote the preferred and less preferred responses given a prompt $x$, respectively.

Several variants of DPO, collectively referred to as DPO-like methods, have been developed to improve performance or address specific issues. For instance, identity preference optimization (IPO) (Azar et al., 2024) aimed to mitigate overfitting in preference learning, DPO-positive (DPOP) (Pal et al., 2024) introduced additional regularization to prevent reward degradation, and Kahneman-Tversky optimization (KTO) (Ethayarajh et al., 2024) incorporated insights from prospect theory to better model human decision-making.

Recent researches have focused on developing reference-free DPO-like methods to eliminate the need for a fixed reference model. Simple preference optimization (SimPO) (Meng et al., 2024) introduced a loss function with length normalization and a reward margin, enabling reference-free optimization while addressing response length control. CPO (constrastive preference optimization) (Xu et al., 2024) showed that when the reference model perfectly aligns with the true data distribution of preferred data, the DPO loss is upper-bounded by a simpler loss function without a reference model by assuming a uniform distribution $U$. The preference part of the CPO loss function is given by:

$$\mathcal{L}_{\text{CPO}}^{Preference}(\pi_\theta) = \mathcal{L}_{\text{DPO}}(\pi_\theta; U) = -\mathbb{E}_{(x,y_w,y_l)\sim\mathcal{D}} \left[ \log \sigma \left( \beta \left( \log \pi_\theta(y_w|x) - \log \pi_\theta(y_l|x) \right) \right) \right]. \quad (2)$$

These DPO-like methods not only enhance computational and memory efficiency but also retain comparable optimization performance to the standard DPO.

Most closely related to our work is SPIN (self-play fine-tuning) (Chen et al., 2024), which employed a self-play mechanism to improve an LLM without additional human preference data iteratively. SPIN generated its training data and refines itself by distinguishing between current and previous outputs, continuously updating its reference model.

Our research bridges the gap between language model alignment and imitation learning, demonstrating how DPO-like alignment techniques, originally developed for text generation problems in LLMs, can be successfully adapted to control problems in reinforcement learning. This cross-domain application opens up new possibilities for improving imitation learning in complex, real-world tasks, and highlights the potential of adapting DPO-like methods.

## 3 PREFERENCE OPTIMIZATION FOR IMITATION LEARNING

### 3.1 ADAPTING DPO-LIKE METHODS FOR IMITATION LEARNING

Our approach leverages the strengths of DPO-like methods, and namely combines the self-play mechanism from SPIN with the reference-free optimization from CPO, to enhance the imitation learning process. In this method, we eliminate the need for a reference model in the self-play setup by adopting CPO's reference-free loss function and allowing us to apply self-play in offline imitation learning without the computational overhead of maintaining a reference model. Specifically, we focus on iteratively refining the agent's policy by directly comparing its actions to those of experts, thus enabling the model to align more closely with expert behavior while reducing computational complexity.

In SPIN, a model generates synthetic data and refines itself by distinguishing between current and previous outputs using a continuously updated reference model. However, maintaining and updating

this reference model adds computational complexity. To address this, we adapt CPO's reference-free optimization, eliminating the need for a reference model while retaining the benefits of self-play.

In our approach, expert demonstrations serve as positive samples ($y_w = a_E$), while the model's own actions during training are treated as negative samples ($y_l = a$), where $a$ denotes the agent's action and $a_E$ denotes the expert's action. This direct comparison enables the model to prioritize actions that align more closely with expert behavior. By progressively learning from its own generated data in relation to expert demonstrations, the agent refines its policy, achieving a more stable and efficient imitation learning process without relying on predefined rewards or reference models.

### 3.2 POIL OBJECTIVE

Our goal is to adapt DPO-like methods to the offline IL setting by eliminating the need for a reference model and incorporating a BC regularization term to enhance performance. Inspired by the findings of CPO as described in Subsection 2.3, we consider the expert's policy as the true data distribution of preferred actions. First, we obtain a reference-free loss function for imitation learning as follows.

$$\mathcal{L}_{\text{POIL}}(\pi_\theta) = -\mathbb{E}_{(s, a_E, a) \sim \mathcal{D}_E} \left[ \log \sigma \left( \beta \left( \log \pi_\theta(a_E|s) - \log \pi_\theta(a|s) \right) \right) \right], \tag{3}$$

This loss function is designed to achieve two main objectives:

1. Align with expert behavior. By maximizing $\log \pi_\theta(a_E|s)$, we encourage the agent to assign higher probabilities to the expert's actions.

2. Discourage sub-optimal actions. By minimizing $\log \pi_\theta(a|s)$, we encourage the agent to move away from sub-optimal (agent's) behaviors.

The loss design of POIL (Equation 3) is to simultaneously minimize the divergence between the expert's behavior and the agent's behavior while maximizing the preference of expert actions over the agent's current actions.

To further encourage the policy to closely mimic expert actions, we incorporate a BC (behavior cloning) regularization term, similar to the approach in Xu et al. (2024). Specifically, we add the negative log-likelihood of expert actions under the agent's policy:

$$\mathcal{L}_{\text{BC}}(\pi_\theta) = -\mathbb{E}_{(s, a_E) \sim \mathcal{D}_E} \left[ \log \pi_\theta(a_E|s) \right]. \tag{4}$$

Our overall augmented POIL loss function then combines the preference optimization and the BC regularization:

$$\mathcal{L}_{\text{POIL}}^{\text{aug}}(\pi_\theta) = \mathcal{L}_{\text{POIL}}(\pi_\theta) + \lambda \cdot \mathcal{L}_{\text{BC}}(\pi_\theta), \tag{5}$$

where $\lambda$ is a hyper-parameter that balances the trade-off between preference optimization and behavior cloning.

By incorporating $\lambda$ as a tunable parameter, we allow for greater flexibility in balancing the influence of the BC regularization term, which is crucial in scenarios with varying quality or quantity of expert data.

Our approach builds upon the proof provided by Xu et al. (2024), adapting it to the offline imitation learning setting and introducing $\lambda$ to enhance the method's adaptability. This results in a loss function that effectively guides the policy towards aligning with the expert data distribution without the need for a reference model.

### 3.3 ALGORITHM

The POIL algorithm, detailed in Algorithm 1, iteratively refines the agent's policy to better align with expert behavior through preference-based optimization and behavior cloning regularization. As shown in Figure 1, the process begins by initializing the policy parameters randomly. During

each iteration, the algorithm samples state-action pairs from the expert demonstrations, which serve as the basis for learning.

To generate agent actions in a differentiable manner, we employ the *reparameterization trick* when sampling from the policy $\pi_\theta(a|s)$. Specifically, the policy outputs a mean $\mu_\theta(s)$ and a standard deviation $\zeta_\theta(s)$. We sample actions using $a = \tanh(\mu_\theta(s) + \zeta_\theta(s) \cdot \epsilon)$, where $\epsilon \sim \mathcal{N}(0, I)$. This approach ensures that gradients flow through the sampling process during optimization, allowing effective updating of the policy parameters $\theta$.

---

**Algorithm 1** Preference Optimization for Imitation Learning (POIL)

---

**Require:** Expert dataset $\mathcal{D}_E = \{(s_i, a_{E,i})\}_{i=1}^N$, scaling factor $\beta$, regularization coefficient $\lambda$, batch size $m$, learning rate $\eta$, number of iterations $T$.
1: Randomly initialize policy parameters $\theta$
2: **for** iteration = 1 to $T$ **do**
3:     Sample a batch of expert state-action pairs $\{(s_j, a_{E,j})\}_{j=1}^m$ from $\mathcal{D}_E$
4:     Generate agent actions $a_j$ via reparameterized sampling from $\pi_\theta(a|s_j)$ for all $j$
5:     **for** each $(s_j, a_{E,j}, a_j)$ in batch **do**
6:         Compute the POIL loss: $\mathcal{L}_{\text{POIL}} = -\log \sigma\left(\beta\left(\log \pi_\theta(a_{E,j}|s_j) - \log \pi_\theta(a_j|s_j)\right)\right)$
7:         Compute the BC regularization term: $\mathcal{L}_{\text{BC}} = -\log \pi_\theta(a_{E,j}|s_j)$
8:         Combine the losses: $\mathcal{L}_{\text{POIL}}^{\text{aug}} = \mathcal{L}_{\text{POIL}} + \lambda \cdot \mathcal{L}_{\text{BC}}$
9:         Update policy parameters: $\theta \leftarrow \theta - \eta \nabla_\theta \mathcal{L}_{\text{POIL}}^{\text{aug}}$
10:     **end for**
11:     **return** $\theta$
12: **end for**

---

In Algorithm 1, $\mathcal{D}_E$ represents the dataset of expert demonstrations, and $N$ is the total number of expert state-action pairs. The regularization coefficient $\lambda$ controls the weight of the BC regularization relative to the preference-based loss, while the scaling factor $\beta$, learning rate $\eta$, and number of iterations $T$ are hyper-parameters that control the optimization process. The batch size $m$ determines how many samples are used in each iteration to compute the gradient.

The algorithm proceeds by sampling batches of expert data and generating corresponding agent actions. The total loss $\mathcal{L}_{\text{total}}$ is computed for each sample in the batch, balancing between preference optimization and behavior cloning. The policy parameters $\theta$ are then updated using gradient descent to minimize the total loss, thereby improving the agent's policy to better match the expert's behavior.

## 4 EXPERIMENTS

In this section, we evaluate the performance of POIL on control tasks in the MuJoCo environment (Todorov et al., 2012). Additionally, to demonstrate the scalability and robustness of POIL in more complex environments, we provide experimental results on the Adroit tasks from the D4RL benchmark (Fu et al., 2020) in Appendix A. We conduct several experiments[1] to demonstrate the effectiveness of POIL under different settings and compare it against some baseline methods, which include those state-of-the-art in the past.

### 4.1 EXPERIMENTAL SETUP

We conduct experiments on standard MuJoCo control tasks, specifically `HalfCheetah-v2`, `Hopper-v2`, and `Walker2d-v2`. These environments are widely used benchmarks in reinforcement learning, requiring agents to learn complex locomotion behaviors in high-dimensional state and action spaces.

For the single demonstration experiments, we utilize one expert trajectory per task, sourced from the same dataset as used in ValueDICE (Kostrikov et al., 2019). These expert trajectories are generated by well-trained policies and present a challenging setting for imitation learning due to the limited data available.

---

[1]Refer to the supplementary to reproduce experiments in this section.

In the experiments involving different dataset sizes, we use datasets from the D4RL benchmark (Fu et al., 2020). We create subsets of the full dataset by randomly selecting different percentages of the data, namely 100%, 10%, 5%, and 2%. This approach allows us to assess the performance of POIL and baseline methods under different data availability conditions.

We compare the performance of POIL against several baseline methods, including BC (Pomerleau, 1991), IQ-Learn (Garg et al., 2021), DMIL (Zhang et al., 2023), and O-DICE (Mao et al., 2024), many of which were state-of-the-art in offline imitation learning in the past. These methods provide a comprehensive benchmark for evaluating POIL.

For fair comparison, we use the same neural network architecture for all methods. The policy network consists of two fully connected layers, each with 256 units, with ReLU activation functions applied after each layer. All models are trained for 100k timesteps. We use the Adam optimizer (Kingma, 2014) for optimization with default parameters. The experiments were conducted on a system equipped with 4 NVIDIA RTX A6000 GPUs, 128GB of RAM, and an AMD Threadripper PRO 5965WX processor featuring 24 cores and 48 threads.

### 4.2 SINGLE DEMONSTRATION LEARNING

In this experiment, we evaluate the ability of POIL to learn effective policies from a single expert trajectory. The hyper-parameter $\beta$ is set to 0.2 in this experiment.

This experiment tests the data efficiency of imitation learning methods when only minimal expert data is available. Detailed results, including standard deviations over multiple runs, are provided in Appendix B.

Table 1: The performance of various methods trained with a single expert demonstration on MuJoCo tasks. The results are averaged over 3 different runs, each using a unique random seed, and the scores represent the average over the last ten epochs. (The bold numbers represent the best, while the underscored numbers are the second best. Note that the scores are not normalized to expert data because we cannot directly get the expert scores from this dataset (Kostrikov et al., 2019)).

| Environment | Traj. | BC | IQ-Learn | DMIL | O-DICE | POIL$_{\lambda=1}$ | POIL$_{\lambda=0}$ |
|---|---|---|---|---|---|---|---|
| HalfCheetah | traj1 | 2775.39 | 3621.73 | 2764.81 | 1532.63 | 3843.75 | **4628.88** |
| | traj2 | 3031.12 | 2957.77 | 3546.40 | 1801.12 | 2228.04 | **4833.13** |
| | traj3 | 2927.38 | 3805.18 | 2802.43 | 1032.74 | 2645.44 | **4309.46** |
| Hopper | traj1 | 1548.32 | 2431.73 | 2947.13 | 1672.45 | 2426.22 | **3501.67** |
| | traj2 | 1687.93 | 2601.54 | **3195.28** | 1748.63 | 2534.35 | 3066.00 |
| | traj3 | 1850.54 | 2520.18 | 3301.41 | 1797.43 | 3372.09 | **3499.08** |
| Walker2d | traj1 | 881.23 | 1011.89 | 1942.76 | 1543.67 | 2645.29 | **4722.79** |
| | traj2 | 1023.85 | 1062.52 | 1686.98 | 1392.54 | 1826.08 | **5233.93** |
| | traj3 | 935.63 | 997.21 | 1864.76 | 1483.52 | 2493.53 | **3745.50** |

As shown in Table 1, POIL with $\lambda = 0$ achieves the highest performance across all trajectories in the `HalfCheetah-v2` task, outperforming all baseline methods. In the Walker2d-v2 task, POIL with $\lambda = 1$ wins on two trajectories, and POIL with $\lambda = 0$ for one trajectory. For the `Hopper-v2` task, POIL with $\lambda = 1$ and $\lambda = 0$ each achieves the best performance on one trajectory, with DMIL outperforming both POIL variants on the remaining one trajectory. Overall, our method only loses to DMIL on one trajectory and outperforms all baseline methods on the remaining tasks. Particularly, POIL with $\lambda = 0$ performs the best on five trajectories, the second on three, and the third on one.

This demonstrates POIL's ability to effectively utilize limited expert data and suggests superior data efficiency compared to other methods. More discussion about $\lambda$ is given inSubsection 4.4.2.

### 4.3 DIFFERENT DATASET SIZES

In this experiment, we assess the performance of POIL and baseline methods when trained on different proportions of the D4RL datasets: 100%, 10%, 5%, and 2%. This evaluation tests the robustness of the methods under different amounts of training data, particularly in data-scarce scenarios.

As shown in Table 2, POIL demonstrates highly competitive performance across different dataset sizes and tasks. In the `HalfCheetah-v2` and `Hopper-v2` tasks, POIL with $\lambda = 1$ performes the best, while POIL $\lambda = 0$ performs the second best for all dataset sizes among all baseline methods. In the `Walker2d-v2` task, POIL with $\lambda = 0$ performs competitively. While IQ-Learn and O-DICE achieve the highest scores in this task, POIL remains very close to them in performance within 1% for all dataset sizes. Particularly for 10% dataset size, it achieves the best result.

Table 2: Performance of different methods trained on varying dataset sizes from D4RL on MuJoCo tasks. Results are normalized to the expert scores, and the scores represent the maximum scores achieved across the training process. (The bold numbers are the best, while the underscored numbers are the second best.)

| Environment | Ratio | BC | IQ-Learn | DMIL | O-DICE | POIL$_{\lambda=1}$ | POIL$_{\lambda=0}$ |
|---|---|---|---|---|---|---|---|
| HalfCheetah | 100% | 91.95 | 80.69 | 94.62 | 95.29 | **96.60** | 95.75 |
| | 10% | 90.64 | 61.48 | 95.37 | 94.67 | **96.14** | 95.42 |
| | 5% | 82.90 | 58.42 | 93.70 | 91.79 | **96.47** | 95.29 |
| | 2% | 23.58 | 44.17 | 89.89 | 92.73 | **95.51** | 95.08 |
| Hopper | 100% | 95.06 | 110.41 | 112.50 | 114.93 | 116.23 | **118.86** |
| | 10% | 83.52 | 104.22 | 112.66 | 114.33 | **117.66** | 117.36 |
| | 5% | 73.35 | 100.13 | 112.93 | 117.29 | 115.45 | **119.01** |
| | 2% | 53.54 | 98.35 | 112.81 | 113.67 | 116.93 | **119.16** |
| Walker2d | 100% | 107.35 | **113.67** | 109.24 | 111.85 | 110.01 | 112.97 |
| | 10% | 105.36 | 111.64 | 110.40 | 110.40 | 110.00 | **112.65** |
| | 5% | 103.21 | **111.84** | 109.53 | 110.79 | 110.15 | 111.19 |
| | 2% | 58.34 | 110.58 | 109.32 | **110.64** | 110.46 | 110.34 |

These results indicate that POIL not only excels in challenging environments like `HalfCheetah-v2` and `Hopper-v2`, but also performs competitively in `Walker2d-v2`. The consistent high performance across different dataset sizes underscores POIL's robustness and effectiveness in offline imitation learning, even in data-scarce scenarios. More discussion about $\lambda$s is given in Subsection 4.4.2.

## 4.4 ABLATION STUDY

### 4.4.1 IMPACT OF SCALING FACTOR $\beta$

We conduct an ablation study to analyze the impact of the scaling factor $\beta$ on POIL's performance. Figure 2 shows the performance of POIL on the `HalfCheetah-v2`, `Hopper-v2`, and `Walker2d-v2` tasks with different values of $\beta$. We set $\lambda$ to zero based on the observations from Subsection 4.2 and Subsection 4.3, where POIL with $\lambda = 0$ demonstrates better performance in most cases.

Our observations indicate that a smaller value of $\beta$, specifically $\beta = 0.2$, yields the best performance across most tasks in this setting. This suggests that choosing an appropriate scaling factor effectively controls the sharpness of the preference function in the loss, influencing how strongly the model distinguishes between expert and agent actions. A smaller $\beta$ value tends to smooth the preference function, which leads to more stable gradients and improved training dynamics, as we will further explore when discussing different choices of $\lambda$ in Subsection 4.4.2.

While $\beta = 0.2$ performed well in most cases as above, it may not universally be the best choice for all tasks or environments. Different environments may require different scaling factors based on their complexity and the nature of the expert data. A consistent observation that smaller $\beta$ values yield better performance suggests that a smoother preference function be generally beneficial in offline imitation learning with POIL.

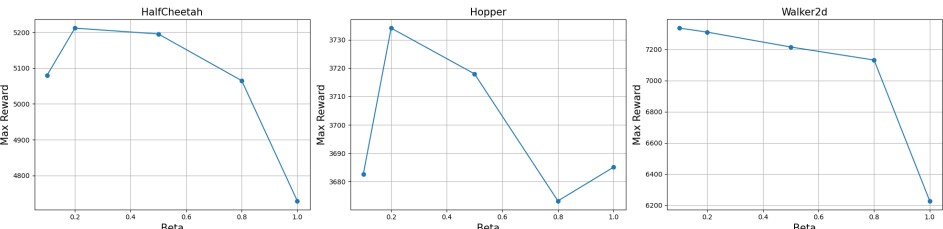

Figure 2: Impact of the scaling factor $\beta$ on POIL's performance in the `HalfCheetah-v2`, `Hopper-v2`, and `Walker2d-v2` tasks.

### 4.4.2 IMPACT OF BC COEFFICIENT $\lambda$

In addition to $\beta$, we also conduct further experiments to analyze the impact of the regularization coefficient $\lambda$, which controls the weight of the BC term. These experiments were conducted under the same conditions as described in Section 4.2, using only a single expert demonstration for each of the three environments, `HalfCheetah-v2`, `Hopper-v2`, and `Walker2d-v2`. The results are averaged over three different random seeds to ensure robust evaluation. Figure 3 illustrates the performance of POIL across the three tasks as $\lambda$ varies from 0 to 3, specifically $\lambda = [0, 0.2, 0.4, 0.6, 0.8, 1.0, 3.0]$, for five different $\beta = [0.1, 0.2, 0.5, 0.8, 1.0]$.

Our findings consistently show that smaller values of $\lambda$ lead to higher performance across all three tasks for most $\beta$. In particular, $\lambda = 0$ yields the highest returns for most $\beta$ in all three environments. As $\lambda$ increases, performance decreases notably, especially for larger values of $\lambda$, such as 3.0, where performance deteriorates significantly across the board.

This suggests that the contribution of the BC regularization term, controlled by $\lambda$, is not as beneficial in this offline imitation learning setup, particularly when smaller $\beta$ values already balance the preference-based loss effectively. Higher values of $\lambda$ seem to introduce a trade-off that limits the agent's ability to align its policy with the expert data, resulting in lower average returns.

However, as demonstrated in Subsection 4.3, when more demonstration data is available, both $\lambda = 0$ and $\lambda = 1$ perform equally well across various dataset sizes, indicating that the regularization term's effect varies depending on the amount of available expert data. In more data-rich environments, moderate values of $\lambda$ tend to help stabilize training without severely hindering performance, unlike in the single demonstration scenario.

These results indicate that POIL performs best with minimal or no BC regularization ($\lambda \approx 0$) in data-scarce settings. In contrast, with larger datasets, the impact of $\lambda$ becomes less pronounced, and both $\lambda = 0$ and $\lambda = 1$ lead to competitive performance.

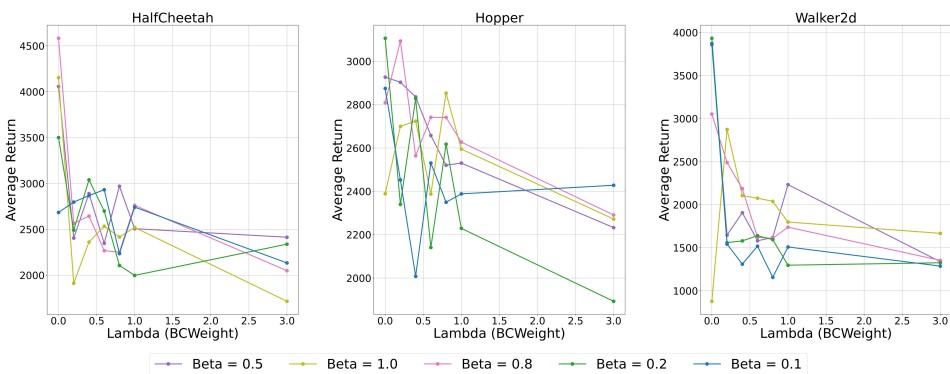

Figure 3: Performance comparison between different $\lambda$ values across various $\beta$ settings on the `HalfCheetah-v2`, `Hopper-v2`, and `Walker2d-v2` tasks.

### 4.4.3 COMPARISON BETWEEN DIFFERENT PREFERENCE OPTIMIZATION METHODS

In POIL, we utilize the CPO loss to achieve good performances in the tasks above. In addition to CPO, there are still many other reference-free preference optimization methods proposed in the context of preference learning in LLM that can be adapted to offline imitation learning, too. In this subsection, we include these methods for comparison, including SimPO (Meng et al., 2024), SLIC-HF (Zhao, 2023), RRHF (Yuan et al., 2023), and ORPO (Hong, 2024), to see whether any of them is better.

As shown in Figure 4, POIL (in olive green) significantly outperforms these methods across all tasks. This suggests that our approach of directly comparing agent actions with expert actions using the POIL loss function is more effective in guiding the policy learning process in offline imitation learning. The superior performance of POIL highlights its effectiveness over other reference-free methods.

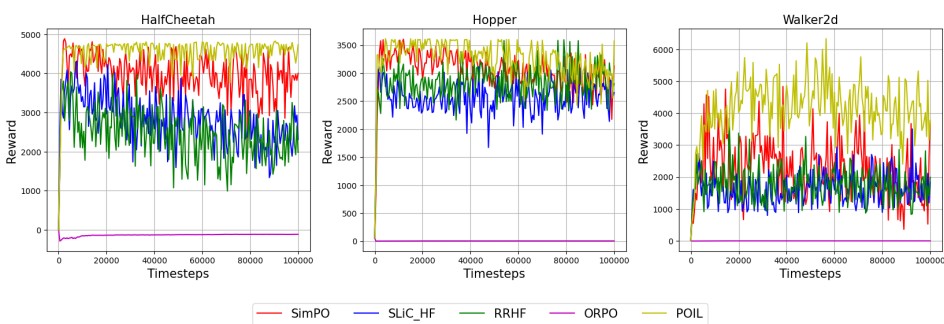

Figure 4: Performance comparison between POIL and other reference-free preference optimization methods on single demonstration task.

## 5 DISCUSSION

In this paper, we introduce POIL, a novel method inspired by preference optimization techniques from large language model alignment. POIL eliminates the need for adversarial training by directly comparing agent actions to expert actions. Through extensive experiments on MuJoCo tasks and the D4RL benchmark, we demonstrate that POIL performs best or competitively against state-of-the-art methods, particularly in data-scarce settings.

A key finding from our study is the role of the regularization coefficient $\lambda$. While CPO typically sets $\lambda = 1$, our results suggest that $\lambda = 0$ often leads to better performance, particularly in simpler tasks or data-scarce settings where overfitting is less of a concern. However, we found that $\lambda$ is more useful in tasks where the expert data is harder to learn from, such as `HalfCheetah-v2`, where the expert reaches a very high score. In such cases, the regularization from $\lambda$ helps the agent stay closer to expert actions, providing a stabilizing effect during training. This allows the agent to learn more effectively in environments where the expert's performance sets a difficult target.

This insight opens potential opportunities for future research. For example, further investigation into tuning $\lambda$ based on task complexity or the quality of the expert data provide more flexible and adaptive learning strategies. In tasks where the expert's performance is exceptionally high, increasing $\lambda$ helps prevent the agent from diverging too far from expert behavior. In contrast, for simpler tasks or when expert data is more limited, reducing $\lambda$ might enable the agent to explore a broader range of policies without over-regularization.

Overall, POIL offers a robust solution for offline imitation learning, especially when expert data is limited or challenging to learn from. Its flexibility in adapting to different dataset sizes and task difficulties makes it a promising direction for future research in imitation learning and related fields.

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

# A    ADDITIONAL EXPERIMENTS ON ADROIT TASKS

To further demonstrate the scalability and robustness of POIL in more complex environments, we conducted additional experiments on the Adroit tasks from the D4RL benchmark (Fu et al., 2020).

## A.1    EXPERIMENTAL SETUP

We focused on the expert datasets within the Adroit tasks, following the same settings as in MCNN (Sridhar et al., 2023). This ensures consistency and allows for a direct comparison between POIL and MCNN under similar conditions. All baseline results reported in our experiments are directly inherited from MCNN. Specifically, we evaluated POIL on the following tasks: `pen-expert-v1`, `hammer-expert-v1`, `door-expert-v1`, and `relocate-expert-v1`. For these experiments, we set the regularization coefficient $\lambda = 0$ and the scaling factor $\beta = 1$ based on our findings from the MuJoCo experiments. We trained our models using the same neural network architecture and optimization settings as in the main experiments to ensure fairness.

## A.2    RESULTS

The performance of POIL on the Adroit tasks is presented in Tables 3 and 4. In Table 3, we compare POIL against the methods reported in the MCNN paper when using the full expert dataset for each task. In Table 4, we evaluate the performance of POIL with varying numbers of demonstrations to assess its data efficiency.

Table 3: Performance comparison on Adroit tasks using the full expert dataset (5,000 demonstrations). The results are averaged over three runs with different random seeds. (The bold numbers indicate the best performance, while the underscored numbers are the second best.)

| Task | BC | BeT-BC | Implicit BC | MCNN (Fixed) | MCNN (Tuned) | POIL |
|------|------|-----------|--------------|----------------|-----------------|----------|
| Pen | 2633 | $1853 \pm 117$ | $2586 \pm 65$ | $3285 \pm 209$ | $3405 \pm 328$ | $\mathbf{4077 \pm 66}$ |
| Hammer | 16140 | $2731 \pm 261$ | $-132 \pm 25$ | $16027 \pm 382$ | $\mathbf{16387 \pm 682}$ | $\underline{16295 \pm 49}$ |
| Door | 969 | $356 \pm 35$ | $361 \pm 67$ | $3033 \pm 0.3$ | $\underline{3035 \pm 7}$ | $\mathbf{3040 \pm 12}$ |
| Relocate | 4289 | $490 \pm 42$ | - | $4566 \pm 47$ | $\underline{4566 \pm 47}$ | $\mathbf{4606 \pm 45}$ |

As shown in Table 3, POIL outperforms all methods reported in the MCNN paper on `pen-expert-v1`, `door-expert-v1`, and `relocate-expert-v1`. On `hammer-expert-v1`, POIL achieves performance comparable to the best method (MCNN+MLP with tuned hyperparameters), with only a slight difference.

Table 4: Performance comparison on Adroit tasks with varying numbers of demonstrations. Results are averaged over three runs with different random seeds. (The bold numbers indicate the best performance for each dataset size, while the underscored numbers are the second best.)

| Task | Demos | MCNN+MLP | POIL |
|------|-------|-------------|----------|
| Door | 100 | $\underline{2725 \pm 139}$ | $\mathbf{3015 \pm 3}$ |
| | 500 | $\underline{2931 \pm 32}$ | $\mathbf{3025 \pm 1}$ |
| | 1000 | $\underline{2992 \pm 18}$ | $\mathbf{3027 \pm 7}$ |
| | 2000 | $\underline{3017 \pm 10}$ | $\mathbf{3028 \pm 22}$ |
| | 4000 | $\underline{3025 \pm 3}$ | $\mathbf{3033 \pm 3}$ |
| | 5000 | $\underline{3035 \pm 7}$ | $\mathbf{3041 \pm 12}$ |
| Pen | 100 | — | $\mathbf{4146 \pm 104}$ |
| | 500 | $\underline{3712 \pm 32}$ | $\mathbf{4127 \pm 8}$ |
| | 1000 | $\underline{3808 \pm 6}$ | $\mathbf{4141 \pm 38}$ |
| | 2000 | $\underline{3858 \pm 29}$ | $\mathbf{4197 \pm 136}$ |
| | 4000 | $\underline{3934 \pm 42}$ | $\mathbf{4172 \pm 113}$ |
| | 5000 | $4051 \pm 195$ | $\mathbf{4078 \pm 66}$ |

Table 4 demonstrates that POIL consistently achieves superior performance across all tasks and dataset sizes. Notably, POIL shows strong performance even with a limited number of demonstrations (e.g., 100 demos), highlighting its data efficiency and robustness in data-scarce scenarios.

# B    ADDITIONAL RESULTS WITH ERROR BARS

In this appendix, we provide detailed results for the single demonstration learning experiments presented in Section 4.2, including standard deviations (error bars) over multiple runs. These results offer a more comprehensive view of the performance variability across different methods.

Table 5: Performance of BC, IQ-Learn, and DMIL trained with a single expert demonstration on MuJoCo tasks. The results are averaged over 3 runs with different random seeds, and the scores represent the average returns ± standard deviation over the last ten epochs. The bold numbers represent the best performance, while the underscored numbers are the second best.

| Environment | Traj. | BC | IQ-Learn | DMIL |
|---|---|---|---|---|
| HalfCheetah | traj1 | 2775.39 ± 296.23 | 3621.73 ± 423.67 | 2764.81 ± 315.29 |
| | traj2 | 3031.12 ± 727.32 | 2957.77 ± 803.21 | 3546.40 ± 521.78 |
| | traj3 | 2927.38 ± 967.15 | 3805.18 ± 567.81 | 2802.43 ± 648.34 |
| Hopper | traj1 | 1548.32 ± 108.74 | 2431.73 ± 698.13 | 2947.13 ± 267.93 |
| | traj2 | 1687.93 ± 70.75 | 2601.54 ± 532.47 | **3195.28 ± 501.47** |
| | traj3 | 1850.54 ± 870.87 | 2520.18 ± 746.91 | 3301.41 ± 687.12 |
| Walker2d | traj1 | 881.23 ± 68.44 | 1011.89 ± 391.27 | 1942.76 ± 321.15 |
| | traj2 | 1023.85 ± 124.71 | 1062.52 ± 287.91 | 1686.98 ± 612.37 |
| | traj3 | 935.63 ± 76.50 | 997.21 ± 159.43 | 1864.76 ± 491.72 |

Table 6: Performance of O-DICE and POIL trained with a single expert demonstration on MuJoCo tasks. The results are averaged over 3 runs with different random seeds, and the scores represent the average returns ± standard deviation over the last ten epochs. The bold numbers represent the best performance, while the underscored numbers are the second best.

| Environment | Traj. | O-DICE | POIL$_{\lambda=1}$ | POIL$_{\lambda=0}$ |
|---|---|---|---|---|
| HalfCheetah | traj1 | 1532.63 ± 716.74 | 3843.75 ± 379.47 | **4628.88 ± 183.47** |
| | traj2 | 1801.12 ± 697.67 | 2228.04 ± 797.59 | **4833.13 ± 30.95** |
| | traj3 | 1032.74 ± 1104.10 | 2645.44 ± 876.50 | **4309.46 ± 247.95** |
| Hopper | traj1 | 1672.45 ± 441.29 | 2426.22 ± 622.98 | **3501.67 ± 469.42** |
| | traj2 | 1748.63 ± 204.60 | 2534.35 ± 398.42 | 3066.00 ± 158.89 |
| | traj3 | 1797.43 ± 145.06 | 3372.09 ± 52.41 | **3499.08 ± 58.08** |
| Walker2d | traj1 | 1543.67 ± 445.67 | 2645.29 ± 840.42 | **4722.79 ± 521.90** |
| | traj2 | 1392.54 ± 1106.87 | 1826.08 ± 490.14 | **5233.93 ± 234.21** |
| | traj3 | 1483.52 ± 716.27 | 2493.53 ± 729.79 | **3745.50 ± 445.90** |

