# OpenReview forum: "POIL: Preference Optimization for Imitation Learning"
_ICLR.cc/2025/Conference — Submitted to ICLR 2025_

### Official Review · Reviewer_kg1L · 2024-10-16

**Soundness:** 3
**Presentation:** 4
**Contribution:** 4
**Rating:** 8
**Confidence:** 3

**Summary:**

The authors introduce a novel algorithm for offline IL, inspired by research from the RLHF literature. The algorithm is quite simple, but demonstrates strong performance against a range of strong baselines.

**Strengths:**

I see this paper as strong on several fronts. It is clearly written and motivated. There are many references to related work. The method is well-explained. There is a solid set of experiments with good baselines and ablations; as far as I can tell, the experimental setup is sound as well.

The final algorithm presented is relatively simple, and I appreciate that the authors do not try to obfuscate this fact. And yet, its performance is apparently quite strong, which should be of significance to the IL community.

**Weaknesses:**

The only criticism that I can produce is that it would be nice to know that this method scales to tasks that are more realistic than the MuJoCo benchmark environments, but the authors have already performed a set of experiments that should be considered sound in this particular research topic (fundamental IL algorithms).

Having tried similar approaches in the past, I was surprised that this method worked as well as it did. But, as shown in the authors' ablations, the scaling factor β is crucial, and in general should be significantly < 1. The authors mention "A smaller β value tends to smooth the preference function, which leads to more stable gradients and improved training dynamics", but do not say/show more about this (most results only show the final return); it would be nice to have more exploration of this.

**Questions:**

- Considering parallels to the LLM literature, I believe that BC is equivalent to supervised fine-tuning (SFT)? An example is in RPO [1]. But it appears that Xu et al. (2024), already cited in the paper, also does this.
- For readers outside of the offline IL literature, how can one determine how long to train for? Are trained policies simply evaluated on the environment, or is it possible to perform some sort of validation?
- As mentioned in the weaknesses, an analysis of how the learned and expert policies change over time could be a good empirical investigation of β's influence on the optimisation process.

[1] Liu, Z., Lu, M., Zhang, S., Liu, B., Guo, H., Yang, Y., ... & Wang, Z. (2024). Provably mitigating overoptimization in rlhf: Your sft loss is implicitly an adversarial regularizer. arXiv preprint arXiv:2405.16436.

---

> ### Author Response · Authors · 2024-11-19
>
> Thank you for your detailed and thoughtful review of our submission. We appreciate your positive assessment of our work and the recognition of the strengths in our approach, experimental design, and presentation.
>
> ### Reply to Weaknesses:
> > The only criticism that I can produce is that it would be nice to know that this method scales to tasks that are more realistic than the MuJoCo benchmark environments, but the authors have already performed a set of experiments that should be considered sound in this particular research topic (fundamental IL algorithms).
>
> Thank you for the comment. For this, we have conducted additional experiments on the **Adroit** tasks from the D4RL benchmark suite, which involve dexterous manipulation using a simulated robotic hand, as shown in the section of "Reply to All Reviewers" (above).
>
> We believe these additional results address your concern and further validate the potential of POIL in practical applications beyond standard benchmarks. We will include these results in the Appendix of our revised paper.
> ___
> > Having tried similar approaches in the past, I was surprised that this method worked as well as it did. But, as shown in the authors' ablations, the scaling factor β is crucial, and in general should be significantly < 1. The authors mention "A smaller β value tends to smooth the preference function, which leads to more stable gradients and improved training dynamics", but do not say/show more about this (most results only show the final return); it would be nice to have more exploration of this.
>
> Thank you for the comment. Indeed, the scaling factor $\beta$ plays a crucial role in POIL's performance. To intuitively illustrate why $\beta$ should be significantly smaller than 1, we provide a simple visualization of the POIL loss function $-\log(\sigma(\beta(\log \pi_\theta(a_E|s) - \log \pi_\theta(a|s))))$.
>
> While we have included a visual representation in the figure (accessible via the link below), OpenReview cannot directly display images.
>
> [Visualization of POIL Loss Function](https://hackmd.io/_uploads/ByYR3CdfJl.png)
>
> The figure demonstrates how the POIL loss varies with different values of $\beta$. The x-axis represents the log likelihood ratio between expert and policy actions. As $\beta$ decreases from 2.0 to 0.1, the loss curve becomes increasingly smooth, leading to more stable gradient signals. This explains our empirical finding that smaller $\beta$ values (typically $<1$ in our experiments) consistently lead to better performance through more stable training dynamics.

---

> > ### Author Response · Authors · 2024-11-19
> >
> > ### Reply to Questions:
> >
> > > 1. Considering parallels to the LLM literature, I believe that BC is equivalent to supervised fine-tuning (SFT)? An example is in RPO [1]. But it appears that Xu et al. (2024), already cited in the paper, also does this.
> >
> > Yes, you are correct in drawing a parallel between Behavior Cloning (BC) in offline IL and Supervised Fine-Tuning (SFT) in the context of Large Language Models (LLMs). In both cases, the objective is to train a model to mimic a set of expert demonstrations using supervised learning. This analogy was a key insight that inspired the development of POIL. We drew upon similarities between techniques in the LLM space (e.g., RLHF and SFT) and imitation learning approaches like GAIL and DAC.
> > Our approach leverages this connection to simplify the training process by using direct preference comparisons without needing a reward model, akin to how SFT optimizes LLMs without requiring explicit adversarial training.
> > ___
> >
> > > 2. For readers outside of the offline IL literature, how can one determine how long to train for? Are trained policies simply evaluated on the environment, or is it possible to perform some sort of validation?
> >
> > Thank you for raising this important question. In our current experiments, we did not set an explicit stopping criterion because we aimed to maintain consistency when comparing POIL with other baseline methods. For a fair comparison, we trained all methods for a fixed number of iterations to standardize the evaluation.
> > However, in real-world scenarios, determining when to stop training can be more nuanced:
> > - **Practical Recommendation**: Ideally, one would periodically evaluate the policy on the actual environment after every few training checkpoints to monitor performance. This approach ensures that the policy does not overfit to the offline dataset and generalizes well to the real environment.
> > - **When Environment Access is Costly or Impractical**: If direct evaluation on the environment is too costly or infeasible (e.g., physical robots or expensive simulations), we recommend using a high-fidelity simulator during training. In such cases, a "sim-to-real" approach can be effective, where the policy is first fine-tuned in simulation and then transferred to the real environment. This can help approximate the optimal stopping point without requiring extensive real-world evaluations.
> >
> > We believe these strategies would help practitioners in applying POIL effectively in real-world applications, balancing training efficiency and deployment costs.
> > ___
> >
> > > 3. As mentioned in the weaknesses, an analysis of how the learned and expert policies change over time could be a good empirical investigation of β's influence on the optimisation process.
> >
> > We agree that a deeper analysis of how the learned policy evolves relative to the expert policy over time could provide valuable insights. Understanding the impact of $\beta$ on this evolution is crucial for explaining why smaller values stabilize training. We will explore adding these analyses in our future work to shed more light on how POIL adapts to varying levels of preference strength during training.

---

> > > ### Author Response · Authors · 2024-11-22
> > >
> > > Thank you for your insightful comment and positive evaluation, which will help us further refine and enhance the quality of our revision. Should there be any additional points requiring clarification or elaboration, we would be more than happy to address them.

---

> > > ### Comment · Reviewer_kg1L · 2024-11-22
> > > **Response to Authors' Update**
> > >
> > > Thank you to the authors for their update - I believe these address any strong concerns I had. Considering the other reviews and the authors' update, I will be keeping my current recommendation (8).

---

> > > > ### Author Response · Authors · 2024-11-27
> > > >
> > > > Thank you very much again. We are glad to hear that our responses have addressed your concerns.

---

### Official Review · Reviewer_3Rvp · 2024-10-31

**Soundness:** 3
**Presentation:** 3
**Contribution:** 3
**Rating:** 8
**Confidence:** 4

**Summary:**

This paper proposes the use of techniques from the preference learning literature for offline imitation learning. The paper utilizes recent techniques from the offline preference-learning literature such as DPO/SPIN/CPO (where the policy's log probability of an action is treated as the reward of the action) and proposes to utilize the combination of a DPO-style loss and a BC loss to prevent overfitting. The authors show surprisingly good results, especially in the low-data regime, where POIL outperforms state-of-the-art model-free offline imitation learning algorithms including IQ-Learn, DMIL, and DICE variants.

**Strengths:**

The experimental section is very well ablated, with strong performance of the proposed method. I am quite surprised that POIL does well in the low-data regime, especially considering most offline preference learning algorithms are very data hungry in the LLM finetuning context (albeit with larger models come larger dataset necessities). The ablation studies of the proposed method were also pretty exhaustive, where the authors ablated over the impact of the preference temperature parameter and the BC loss they use in practice.

**Weaknesses:**

There is no explicit theoretical justification in this work, but this is minor to me. I think DPO and its variants have strong theory as is when it comes to solving the KL-regularized RL problem. I feel like with less than 1 expert trajectory (which some offline imitation learning methods look into, albeit with additional suboptimal data) the method fails, but maybe this is not necessary in real-world settings.

I am curious as to whether an RLHF-centric approach to this problem can be good to compare to (e.g. in LLMs there is the PPO w/ RM vs. DPO debate), where one trains a reward model on policy data (low reward) vs expert data (high reward) as the agent trains. In some sense, I feel like this method is similar to DAC, which is outperformed by IQ-Learn anyway, so maybe this is unnecessary.

**Questions:**

No questions from my perspective for now.

---

> ### Author Response · Authors · 2024-11-19
>
> Thank you for your positive review and your valuable feedback on our submission. We are pleased that you found the experimental results compelling, particularly our method’s performance in the low-data regime.
>
> ### Reply to Weaknesses
> > There is no explicit theoretical justification in this work, but this is minor to me. I think DPO and its variants have strong theory as is when it comes to solving the KL-regularized RL problem.
>
> Thank you for the question. We have addressed some theoretical aspects in our response to Reviewer xCF9. In that response, we provide mathematical insights in more detail into the underpinnings of POIL, elaborating on its relationship with Behavioral Cloning (BC) and preference-based methods.
>
>
> ___
>
> > I feel like with less than 1 expert trajectory (which some offline imitation learning methods look into, albeit with additional suboptimal data) the method fails, but maybe this is not necessary in real-world settings.
>
> You are correct that our current experiments did not evaluate the scenario with less than one expert trajectory. Our focus was primarily on settings where a moderate amount of expert data is available, as we believe this reflects many practical real-world applications where collecting at least some expert demonstrations is feasible.
>
> However, we agree that exploring the performance of POIL in ultra-low-data regimes is an important direction for future research. In scenarios with less than one expert trajectory, methods that can effectively leverage additional suboptimal data or incorporate data augmentation techniques might be necessary to achieve satisfactory performance. While POIL in its current form may face challenges in such settings, we are optimistic that extensions of our method could address these challenges.
>
>
>
> ___
>
> > I am curious as to whether an RLHF-centric approach to this problem can be good to compare to (e.g. in LLMs there is the PPO w/ RM vs. DPO debate), where one trains a reward model on policy data (low reward) vs expert data (high reward) as the agent trains. In some sense, I feel like this method is similar to DAC, which is outperformed by IQ-Learn anyway, so maybe this is unnecessary.
>
> Your suggestion to explore an RLHF-centric approach is insightful, especially considering the similarities between preference learning in RL and LLM contexts. While RLHF typically requires training a reward model, POIL avoids this by using direct preference comparisons, which simplifies the process in offline settings where expert data is limited. We see this as a promising direction for future research and appreciate your suggestion.
>
> We appreciate your insightful observation regarding the similarities between RLHF (Reinforcement Learning from Human Feedback) approaches and traditional imitation learning techniques like DAC (Discriminator Actor-Critic). In fact, this connection was one of the key inspirations for developing POIL.
>
> Your recognition of this connection highlights a core insight that drove the development of POIL. By leveraging preference optimization without the overhead of adversarial training or online interactions, POIL is able to achieve efficient and effective imitation learning, particularly in data-limited scenarios.
> ___

---

> > ### Author Response · Authors · 2024-11-22
> >
> > Thank you for your insightful comment and positive evaluation, which will help us further refine and enhance the quality of our revision. Should there be any additional points requiring clarification or elaboration, we would be more than happy to address them.

---

> > > ### Comment · Reviewer_3Rvp · 2024-11-26
> > >
> > > Hi,
> > >
> > > Thanks for the comments, and very sorry for the late response. The responses here have addressed my primary concerns well. Thanks!

---

> > > > ### Author Response · Authors · 2024-11-27
> > > >
> > > > Thank you very much again. We are glad to hear that our responses have addressed your concerns.

---

### Official Review · Reviewer_xCF9 · 2024-11-03

**Soundness:** 2
**Presentation:** 3
**Contribution:** 1
**Rating:** 3
**Confidence:** 5

**Summary:**

The authors consider framing offline IL as a preference learning problem. Specifically, they generate actions from the learner on states from the expert demonstrations and try to raise the relative probability of the expert actions to the learner actions.

**Strengths:**

-  A generally clear exposition of the proposed method.

**Weaknesses:**

Apologies for LaTex failing to render properly below -- I spent some time playing around with things to no avail.

I am fairly confident the proposed method is provably equivalent to a noisy version of behavioral cloning. To see this, first note that for policies in the exponential family (e.g. Gaussians), we can always write $\pi(a|s) = \exp(f(s, a)) / \sum_{a'} \exp(f(s, a'))$. Then, the likelihood gradient can be expanded to $\nabla_{\theta} \log \pi_{\theta}(a_E|s) = \nabla_{\theta} f_{\theta}(s, a_E) - \nabla_{\theta} \log \sum_{a'} \exp(f_{\theta}(s, a')) = \nabla_{\theta} f_{\theta}(s, a_E)  - \mathbb{E}_{a \sim \pi_{\theta}(\cdot|s)}[\nabla_{\theta} f_{\theta}(s, a)]$. Observe that if we ignore the $\log \sigma$ in the POIL loss for a moment, the BC gradient is simply the "infinite sample" estimate of POIL loss. Put differently: MLE in exponential families already includes a "negative gradient," there is no need to add one in explicitly.

For the specific case of Gaussians, for which the sufficient statistics / moments ($f_{\theta}$ in the above notation) are the mean and variance, the negative gradient term is basically computing the mean by sampling from the policy. I don't see how this can provide any value compared to straight MLE / BC where we just use the fact we know the mean because we know the policy.

Of course, one might then ask about the effect of the $\log \sigma$. If one recalls the original MaxEnt IRL / DPO derivations, this term in the loss is meant to ensure closeness to the prior / reference policy. However, the POIL loss does not include any regularization to the prior (i.e. there is no $\pi_{ref}$ in the denominator), so this is at best providing entropy regularization (i.e. bumping up the variance for a Gaussian policy). This could be done without any samples from the policy, which again begs the question of what we're getting out of this more complex procedure compared to BC.

**Questions:**

1. Could you please add standard error bars to all the plots / tables in the paper?

2. Could you test out your method on something other than the three easiest Mujoco environments?

3. Could you try the "linear" version of the loss I discuss in the weaknesses section? Maybe try tacking on a variance bonus?

---

> ### Author Response · Authors · 2024-11-19
>
> Thank you for your mathematical analysis and the theoretical connections you've drawn between POIL and BC. Your analysis prompts us to better explain these technical aspects in depth. We address your concerns point by point below:
>
> ### Reply to Weaknesses
> > If we ignore the term involving $(\log \sigma)$ in the POIL loss for a moment, the Behavioral Cloning (BC) gradient is simply the "infinite sample" estimate of the POIL loss. In other words, Maximum Likelihood Estimation (MLE) in exponential families already includes a "negative gradient," so there is no need to add one explicitly.
>
> To elaborate on the relationship between POIL and BC, we would like to share several key observations:
>
> First, a lot of simulation environments, e.g. Adroit Hand (Door, Hammer, etc.), Gym-Mujoco (HalfCheetah, Walker2d, etc.), Franka Kitchen, Ant Maze, and so on, constrain action spaces to $[-1, 1]$. Generally, to handle such bounded action spaces, people would use tanh to map unbounded Gaussian samples to bounded actions (squashed Gaussian), as demonstrated in SAC (Soft Actor-Critic) [SAC] and f-IRL (Inverse Reinforcement Learning via State Marginal Matching) [f-IRL]. Compared to direct clipping actions which distorts the Gaussian density at boundaries, this approach maintains a well-defined probability distribution. Specifically, an action $a$ is generated by:
> $$a = \tanh(a_{in}), \text{ where } a_{in} \sim \mathcal{N}(\mu_\theta(s), \sigma_\theta(s))$$
>
> As detailed in SAC's Appendix C [SAC], this transformation introduces a probability correction term $-\log(1-\tanh^2(a_{in}))$ in the log-likelihood due to the change of variables.
>
> For basic BC with squashed Gaussian policy, the objective is:
> $$
> L\_{BC}(\theta) = -\mathbb{E}\_{(s,a\_E)\sim D\_E}[\log \pi\_\theta(a\_E|s)]
> $$
> $$
> = -\mathbb{E}\_{(s,a_E)\sim D\_E}[\log \mathcal{N}(a\_{in,E}; \mu\_\theta(s), \sigma\_\theta(s)) - \log(1-\tanh^2(a\_{in,E}))]
> $$
> $$
> = -\mathbb{E}\_{(s,a_E)\sim D\_E}[-\frac{1}{2}\log(2\pi\sigma\_\theta(s)^2) - \frac{(a\_{in,E}-\mu\_\theta(s))^2}{2\sigma\_\theta(s)^2} - \log(1-\tanh^2(a\_{in,E}))]
> $$
>
> The gradients are:
> $$
> \nabla\_\mu L\_{BC} = \mathbb{E}\_{(s,a\_E)\sim D\_E}\bigg[\frac{a\_{in,E} - \mu\_\theta(s)}{\sigma\_\theta(s)^2}\bigg] = \mathbb{E}\_{(s,a\_E)\sim D\_E}[g\_\mu]
> $$
> $$
> \nabla\_\sigma L\_{BC} = \mathbb{E}\_{(s,a\_E)\sim D\_E}\bigg[-\frac{1}{\sigma\_\theta(s)} + \frac{(a\_{in,E} - \mu\_\theta(s))^2}{\sigma\_\theta(s)^3}\bigg] = \mathbb{E}\_{(s,a\_E)\sim D\_E}[g\_\sigma]
> $$
>
> Let $g_\mu$ and $g_\sigma$ denote the terms inside the expectations of BC gradients. We would use it later.
>
> While the probability correction term $-\log(1-\tanh^2(a_{in,E}))$ appears in the objective, it does not affect the gradients in BC since it doesn't depend on the policy parameters $\theta$. The key issue lies in how BC handles the pre-tanh space directly: when the same action space correction is needed, the required parameter updates vary dramatically based on position in the action space. To illustrate this:
>
> Consider two cases where we want the same 0.001 correction in action space:
> 1. Center: $μ = 0.0, a_E = 0.001$
>    - In action space: $\tanh(0.0) = 0.0$, $a_E = 0.001$
>    - Required update in pre-tanh space: $\tanh^{-1}(0.001) - 0.0 = 0.001$
>
> 2. Near-boundary: $μ = 1.832, a_E = 0.951$
>    - In action space: $\tanh(1.832) = 0.95$, $a_E = 0.951$
>    - Required update in pre-tanh space: $\tanh^{-1}(0.951) - 1.832 = 0.010358$
>
> This analysis shows that while both cases need the same 0.001 correction in action space, the required update in pre-tanh space near boundaries is over 10 times larger than in the center region. This difference arises from the non-linear nature of the tanh transformation: near boundaries (|a| ≈ ±1), the derivative of tanh approaches zero, requiring much larger pre-tanh updates to achieve the same action space correction compared to central regions (|a| ≈ 0) where the derivative is close to one. Since BC operates directly on the pre-tanh parameters, this leads to disproportionately large parameter updates near boundaries. More critically, even small adjustments in action space near boundaries demand substantially larger parameter updates, which can cause potential oscillations and instability during training - the policy might overshoot the target behavior near boundaries and require multiple corrective updates, leading to inefficient learning and potential convergence issues.
>
> In contrast, POIL's sampling-based approach naturally handles these non-linear effects of the tanh transformation, providing more appropriate parameter updates across the entire action space.

---

> ### Author Response · Authors · 2024-11-19
>
> ___
> > One might then ask about the effect of $(\log \sigma)$. If we refer to the original MaxEnt IRL/DPO derivations, this term in the loss is meant to ensure closeness to the prior/reference policy. However, the POIL loss does not include any regularization toward the prior (i.e., there is no $\pi_{\text{ref}}$ in the denominator). Thus, at best, it acts as entropy regularization (i.e., increasing the variance for a Gaussian policy). This could be achieved without requiring any samples from the policy, which raises the question of what additional benefit this more complex procedure provides compared to Behavioral Cloning.
>
> Regarding the reviewer's point about log sigmoid potentially serving only as variance regularization, we analyze BC with log sigmoid:
> $$
> L\_{BC-LS}(\theta) = -\mathbb{E}\_{(s,a\_E)\sim D\_E}\big[\log \sigma(\log \pi\_\theta(a\_E \mid s))\big]
> $$
> $$
> = -\mathbb{E}\_{(s,a\_E)\sim D\_E}\big[\log \sigma(\log \mathcal{N}(a\_{in,E}; \mu\_\theta(s), \sigma\_\theta(s)) - \log(1 - \tanh^2(a\_{in,E})))\big]
> $$
> Let $g_\mu$ and $g_\sigma$ denote the terms inside the expectations of BC gradients as defined above. Then for BC with log sigmoid:
> $$
> \nabla\_\mu L\_{BC-LS} = -\mathbb{E}\_{(s,a\_E)\sim D\_E}\big[(1 - \sigma(\log \mathcal{N}(a\_{in,E}; \mu\_\theta(s), \sigma\_\theta(s)) - \log(1 - \tanh^2(a\_{in,E})))) \cdot g\_\mu\big]
> $$
> $$
> \nabla\_\sigma L\_{BC-LS} = -\mathbb{E}\_{(s,a\_E)\sim D\_E}\big[(1 - \sigma(\log \mathcal{N}(a\_{in,E}; \mu\_\theta(s), \sigma\_\theta(s)) - \log(1 - \tanh^2(a\_{in,E})))) \cdot g\_\sigma\big]
> $$
> Let's examine two cases (same as mentioned above) where policy mean $\mu$ and expert action $a_E$ differ by approximately 0.001 in action space (assuming $\sigma_\theta(s) = 1$ for simplicity):
> 1. Center: $\mu = 0.0$, $a_E = 0.001$
>    - Total log probability: $-0.919$
>    - Weight term = $1 - \sigma(-0.919) \approx 0.715$
>
> 2. Near-boundary: $\mu = 1.832$, $a_E = 0.951$
>    - Total log probability: $1.429$
>    - Weight term = $1 - \sigma(1.429) \approx 0.193$
>
> While BC-LS introduces a weighting mechanism that scales down the large pre-tanh updates near boundaries (weight ≈ 0.193 vs 0.715 in center), the policy still computes gradient updates based on Maximum Likelihood Estimation (MLE) in the pre-tanh space, where updates near boundaries are over 10 times larger (0.010 vs 0.001) than in the center region for the same action space correction. Although the weight term partially mitigates the instability by reducing the effective update size, it does not address the fundamental issue that the policy optimization still operates in a space requiring disproportionately large updates near boundaries.
>
> In contrast, POIL approaches this challenge through a fundamentally different mechanism:
> $$
> L\_{POIL}(\theta) = -\mathbb{E}\_{(s,a\_E)\sim D\_E}\big[\log \sigma(\beta(\log \pi\_\theta(a\_E \mid s) - \log \pi\_\theta(a \mid s)))\big] ,
> \text{where } a \sim \pi\_\theta(\cdot \mid s)
> $$
> The key distinction lies in how POIL handles action probabilities. While both POIL and BC-LS use a $(1-\sigma(\cdot))$ term that moderates updates, POIL's probability ratio approach compares expert actions with sampled actions from the current policy, allowing it to assess the relative preferences across the entire action distribution rather than just at the mean. This leads to partial cancellation of the Jacobian terms ($\log(1-\tanh^2(\cdot))$) between expert and sampled actions, reducing their impact. Additionally, only the Jacobian terms from sampled actions contribute to the gradient computation, providing a more balanced approach to handling bounded actions compared to BC-LS's direct compensation mechanism.

---

> > ### Author Response · Authors · 2024-11-19
> >
> > **Deterministic Version Without Sampling**
> >
> > To address the question of whether sampling is truly necessary for our approach ("This could be achieved without requiring any samples from the policy"), we conducted some initial experiments using a **deterministic version without sampling** that directly uses the mean action:
> > $$
> > L\_{det}(\theta) = -\mathbb{E}\_{(s,a\_E)\sim D\_E}\big[\log \sigma(\beta(\log \pi\_\theta(a\_E \mid s) - \log \pi\_\theta(\mu\_\theta(s) \mid s)))\big]
> > $$
> >
> > These experiments were performed under the same settings as Single Demostration Experiment in our paper, with a training duration of 100,000 steps. Without sampling, we cannot successfully train the model, namely, the performance keep being close to zero over all three tasks, e.g., 20-30 for `Hopper`, 40–50 for `Walker2d`, and -2 to -1 for `Half-Cheetah`. Although this does not definitively prove that the deterministic approach is ineffective, sampling does benefit in achieving high performance.
> >
> > Our use of sampling is actually inspired by techniques in SPIN (Self-Play Fine-Tuning), which aligns with findings in language modeling where deterministic approaches, such as using argmax for next token prediction, often result in repetitive and suboptimal outputs, as demonstrated in [1] (see Page 6).
> >
> >
> > [1] Welleck, S., Kulikov, I., Roller, S., Dinan, E., Cho, K., & Weston, J. (2019). Neural text generation with unlikelihood training. arXiv preprint arXiv:1908.04319.

---

> ### Author Response · Authors · 2024-11-19
>
> ### Reply to Questions
> > 1. Could you please add standard error bars to all the plots / tables in the paper?
>
>
>
> For Table 1 (in this paper), we will provide the result with standard error bars in the next revision.
>
>
> For Table 2 (in this paper), as mentioned in the paper, we report the maximum scores achieved across the training process, normalized to expert performance. Since these are best-case results rather than averages, standard errors/deviations are not applicable in this context.
>
> ___
> > 2. Could you test out your method on something other than the three easiest Mujoco environments?
>
> Thank you for this suggestion. In response, we have conducted additional experiments on the **Adroit** tasks from the D4RL benchmark suite, which involve complex, high-dimensional robotic manipulation tasks using a simulated robotic hand, as shown in the section of "Reply to All Reviewers" (above).
>
> ___
> > 3. Could you try the "linear" version of the loss I discuss in the weaknesses section? Maybe try tacking on a variance bonus?
>
> As we discussed in our response to your weaknesses section, the use of the log-sigmoid function in the POIL loss is crucial for stabilizing gradients and effectively handling the non-linearities introduced by the tanh transformation in squashed Gaussian policies.
>
> We acknowledge that exploring the linear version and assessing its impact on performance is a valuable direction for future research.

---

> > ### Author Response · Authors · 2024-11-22
> >
> > Thank you for your detailed review and feedback. We hope our responses have effectively addressed your questions and concerns. If there are any remaining aspects requiring clarification, we would be happy to provide further details. We trust that the additional information has clarified the strengths of our work, and we hope this may lead to a favorable reconsideration.

---

> > > ### Comment · Reviewer_xCF9 · 2024-11-24
> > > **Re:**
> > >
> > > Hi,
> > >
> > > Apologies for the slow response on my part, crazy week. Thank-you for making several of my suggested changes, providing a detailed rebuttal of some of my comments, and in general engaging in the review process. It is much appreciated. Responding in turn:
> > >
> > > Re: BC vs. POIL. I appreciate the careful gradient tracing here. I don't think this explanation holds water, unfortunately. Let's focus first on the version without the $\log \sigma$ or the squashing. So, if we write out the infinite sample version of the POIL loss (i.e. we subtract a copy of the gradient where we replace all $a_E$ with $a \sim \pi(s)$ in your above expressions for $\nabla_{\mu} L_{BC}$ and $\nabla_{\mu} L_{BC}$), observe that these "negative gradient" terms are **literally** zero. Like, $\mathbb{E}_{s \sim D_E, a \sim \pi(s)}[(a - \mu(s)) / \sigma^2(s)] = 0$. The same is true for $\sigma$. So, if you sampled enough, you would **exactly** get the BC gradient. Thus, in this simplified setting, I don't think there is any reason to believe POIL should out-perform BC.
> > >
> > > Ok, let's introduce the squashing now. Observe that if I sampled some set of actions from the current learner policy, there is no gradient between these sampled actions and the mean. So, if as you say above, the log det Jacobian term doesn't affect the BC loss, I don't see why the log det Jacobian term would affect the linear POIL loss.
> > >
> > > So, taking the above two points together, I would argue that POIL should at best match the performance of BC without the $\log \sigma$ -- the "negative samples" / core algorithmic change on top of base BC give you nothing without it. While I do indeed believe that adding in an arbitrary nonlinearity would make the gradients different, it need not make things better (e.g. some of the ablations in Fig. 4 underperform BC). So, I am still uncomfortable with recommending this paper for acceptance. Sorry :/
> > >
> > > Separately, it might be a good idea to add in the $\log \sigma$ BC you discuss above as a baseline to your paper if you have time.
> > >
> > > Re: reporting maximum scores. Thank-you for bringing this to my attention, I did not catch this on my initial read. Unfortunately, this is not standard practice, at least in imitation learning. Doing this can completely obscure any training instabilities of the method and is a huge red flag. Furthermore, given one needs online interaction to do the rollouts to select the best-performing checkpoint, this significantly detracts from the purported "offline" nature of the method. I would **strongly** suggest reporting the performance of the last checkpoint / using a purely offline model selection procedure (e.g. picking the minimum validation error).

---

> ### Author Response · Authors · 2024-11-27
>
> ### Reply to the Second Comment
>
>
> **1. Differentiable Sampling via Reparameterization Trick**
> > Observe that if I sampled some set of actions from the current learner policy, there is no gradient between these sampled actions and the mean. So, if as you say above, the log det Jacobian term doesn't affect the BC loss, I don't see why the log det Jacobian term would affect the linear POIL loss.
>
> **In POIL, we sample actions from the current policy using the common reparameterization trick, which ensures that these sampled actions are differentiable with respect to the policy parameters $\theta$.** Specifically, the sampling process is:
>
>
>
> $$
> \text{Sample  } \epsilon \sim \mathcal{N}(0, I),
> $$
> $$
> a_{\text{raw}} = \mu_\theta(s) + \zeta_\theta(s) \cdot \epsilon,
> $$
> $$
> a = \tanh(a_{\text{raw}}).
> $$
>
>
> This means that the sampled action $a$ depends on $\theta$ not just through the mean $\mu_\theta(s)$ and standard deviation $\zeta_\theta(s)$, but also through the sampled noise $\epsilon$. In order to make it clearer, we will mention this in our next revision.
>
> ###### Note: To avoid confusion between the standard deviation (previously denoted as $\sigma$) and the sigmoid function $\sigma(\cdot)$ used in the loss function, we now use $\zeta_\theta(s)$ as the symbol for the standard deviation, instead.
>
> **2. Impact of the Log-Determinant Jacobian Term**
> > So, taking the above two points together, I would argue that POIL should at best match the performance of BC without the $\log \sigma$ -- the 'negative samples' / core algorithmic change on top of base BC give you nothing without it.
>
> **The log-determinant Jacobian term, $\log(1 - \tanh^2(a_{\text{raw}}))$, contributes to the gradient in POIL but not in BC, leading to different update directions.** While it is true that this term does not affect the gradient when evaluating at the expert action $a_E$ (since $a_E$ is fixed), it does affect the gradient when evaluating at the sampled action $a$, because $a_{\text{raw}}$ depends on $\theta$.
>
>
> **3. Regarding Ablations in Figure 4 and Performance Relative to BC**
>
> > "While I do indeed believe that adding in an arbitrary nonlinearity would make the gradients different, it need not make things better (e.g. some of the ablations in Fig. 4 underperform BC)."
>
> First of all, we would like to clarify that **the ablation study presented in Figure 4 is a comparison of alternative preference learning methods** (SimPO, SLiC_HF, RRHF, ORPO, POIL), and **our POIL (*olive-green* in the figure) clearly outperform others (definitely NOT UNDERPERFORM)**.
>
>
> Through these experiments, we selected CPO-based loss for POIL because it consistently outperforms others. This indicates that **the specific nonlinearity chosen in POIL is not arbitrary but rather empirically validated as effective.**
>
> **4. Sufficient Sample Argument**
> > So, if you sampled enough, you would exactly get the BC gradient. Thus, in this simplified setting, I don't think there is any reason to believe POIL should out-perform BC.
>
> **Your analysis focuses on a simplified setting without log sigmoid, which fundamentally differs from our method**. Our previous response (in the first reply to BC-LS as well as item one in this reply) has shown that POIL is different from BC.
>
> **In all of our experiments, POIL consistently outperforms BC across all benchmarks**, including the extra one for Adroit tasks (presented in the section of "Reply to All Reviewers")
>
> From above, both theoretical and empirical evidences show that **POIL outperforms BC.**
>
> **5. Reporting Maximum Scores**
> > Re: reporting maximum scores. Thank-you for bringing this to my attention, I did not catch this on my initial read. Unfortunately, this is not standard practice, at least in imitation learning. Doing this can completely obscure any training instabilities of the method and is a huge red flag. Furthermore, given one needs online interaction to do the rollouts to select the best-performing checkpoint, this significantly detracts from the purported "offline" nature of the method. I would strongly suggest reporting the performance of the last checkpoint / using a purely offline model selection procedure (e.g. picking the minimum validation error).
>
> Thank you for requesting. However, since this requires additional experiments for a purely offline model selection procedure and checkpoints, we are not able to provide these soon. In any case, we would like to emphasize that **in all of our experiments, including the newly added Adroit benchmarks, POIL consistently outperforms other methods.** This robust performance across multiple datasets and benchmarks reinforces the effectiveness of POIL.

---

> > ### Comment · Reviewer_xCF9 · 2024-11-28
> >
> > Hi,
> >
> > Sorry, let me make sure I'm understanding this correctly: you don't "detach" the action tensor from the computation graph before throwing it into the POIL loss -- like, $\partial a / \partial \theta \neq 0$? Also, when you're calculating the likelihood, ($\log \pi_{\theta}(\cdot|s)$), do you pass in $a_{raw}$ or $a$? I'm trying to see whether $\theta$ cancels out in the gradient calculation.
> >
> > Relatedly, have you tried the "detached" version where you treat the learner samples as another "BC" dataset, but one you want to move away from? This is the way things are actually implemented in SPIN / online DPO, where there is no gradient w.r.t. the parameters of the model of the learner samples.

---

> ### Author Response · Authors · 2024-11-29
>
> >Sorry, let me make sure I'm understanding this correctly: you don't "detach" the action tensor from the computation graph before throwing it into the POIL loss -- like,$\frac{\partial a}{\partial \theta}\neq 0$?
>
> Yes, we use PyTorch's `torch.rsample()`, which implements the reparameterization trick we described above. You may refer to the official documentation [here](https://pytorch.org/docs/stable/distributions.html#pathwise-derivative)  for details. Through reparameterization, sampled actions maintain differentiable paths with respect to policy parameters.
>
>
>
>
>
> > Also, when you're calculating the likelihood, $(log(\pi_{\theta}(\cdot|s)))$, do you pass in $a_{raw}$ or $a$? I'm trying to see whether cancels out in the gradient calculation.
>
>
> When calculating the likelihood $\log(\pi_{\theta}(a|s))$ under squashed Gaussain policy, we use $a_{\text{raw}}$. Specifically, we compute:
>
> $$
> \log(\pi_{\theta}(a|s)) = \log \mathcal{N}(a_{\text{raw}}; \mu_{\theta}(s), \zeta_{\theta}(s)^2) - \sum_i \log\left(1 - \tanh^2(a_{\text{raw}, i})\right)
> $$
>
> Here, $a_{\text{raw}}$ is used in the Gaussian log-probability and the Jacobian adjustment for the squashing function, while $a = \tanh(a_{\text{raw}})$ is the action applied in the environment.
>
>
> >Relatedly, have you tried the "detached" version where you treat the learner samples as another "BC" dataset, but one you want to move away from? This is the way things are actually implemented in SPIN / online DPO, where there is no gradient w.r.t. the parameters of the model of the learner samples.
>
>
> We have not tried the "detached" version. In our implementation, we allowed gradients to flow through the learner samples back to the policy parameters, the same as in SAC's squashed gaussian policy. However, exploring the "detached" version is an interesting idea, and we will consider it in future work.

---

> > ### Comment · Reviewer_xCF9 · 2024-11-29
> >
> > Hi,
> >
> > Thanks for the above clarification re:rsample.
> >
> > Re: using $a_{raw}$: so, ignoring the $\tanh$ for a moment, we can write out the Gaussian likelihood to get $-\log(\sqrt{2 \pi \zeta(s)^2}) + (-(\epsilon \zeta(s) + \mu(s))^2 / (2\zeta(s)^2)) = -\log(\sqrt{2 \pi \zeta(s)^2}) -\epsilon^2/2$. The second term would vanish when taking a gradient, so we can just focus on the first, also known as the gradient of the log partition function. Now, because the partition function doesn't depend on the action chosen, it should cancel out when subtracting the BC log prob from the learner action log prob. This is why folks like the Bradley-Terry model DPO/SPIN are based on -- one does not need to estimate the partition function like in MaxEnt IRL because the it cancels out in the likelihood calculation. So, I think you'd be left with something like $-(a_E - \mu(s))^2 / (2\zeta(s)^2))$ as the likelihood difference. Now, this objective would incentivize the learner to "blow up" $\zeta(s)$. I would think the reason this doesn't happen is because of the $\tanh()$ penalty, which is serving a role similar to an entropy bonus if I'm not making a sign error.
> >
> > Ok so put together, I would suggest the following changes to the paper in light of our above discussion:
> > 1. First, acknowledge that if we think of the linear version of the loss without the reparameterization trick / log det Jacobian correction, one can at best hope to do as well as BC. You can point out that in theory if one samples enough, they will literally get the BC gradient. You've already worked out the math required for this in the Gaussian setting above. I would suggest also implementing this on some environment to confirm this fact empirically.
> > 2. You can then point out that your "key contribution" is a series of design decisions to unlock the benefits of learner-generated negative samples that the naive approach described above is unable to do. Being honest here will only make your work seem more well thought-out / nuanced / interesting: point 1. can be spun as a "feature" not a '"bug" for your paper.
> > 3. I would begin by observing that in contrast to the way people implement things for SPIN / online DPO, you keep the gradients flow to the learner generated samples. This is the first key difference that us leading to something different than the BC gradient as I observed above. I would strongly suggest independently ablating this change independently on top of the "linear" loss.
> > 4. I would then observe that drawing upon the rich literature on direct alignment algorithms, another way to differentiate yourself from BC is by adding in a nonlinearity on top of the likelihood difference (any monotonic link function should be ok in theory, one of which is $\sigma$). I would suggest applying each of these "link functions" to BC, POIL without rsample, and POIL with rsample. I would expect the first two options to perform similarly. You can then argue that it is the combination of these two changes that leads to POIL's improvement in performance. I don't think merely pointing out the gradients, as you did above, is sufficient here -- we'd need to see empirically how much of a performance improvement there is.
> > 5. You can then pick the best link function and scale things up across multiple environments, reporting either the last or minimum validation error rather than the maximum over training, which can hide a variety of instabilities. More D4RL environments could be  good to look into here. Given the details on which nonlinearity one chooses / the use of rsample matter here (i.e. there's no clear theory reason one should be better than another AFAIK), I think the burden of proof is on the authors to show empirically that their choices scales to multiple harder environments.
> >
> > So, in summary, I think I'm less concerned than when I first read the paper that POIL is secretly BC. I thank the authors for their incredible responsiveness and patience with my many questions, which helped me arrive at this changed viewpoint. That said, I don't think the paper currently does a good job of carefully ablating what actually differentiates POIL from BC: independently, the negative samples do nothing, one necessarily has to combine them with things like rsample and nonlinearities to see a difference (which could be good or bad). Furthermore, I do not think it is acceptable to report results where one takes the max over training in IL, so it is hard for me to give points for the current set of results in Table 2. This makes it difficult for me to fully believe the two differentiating factors from BC are generally a good idea. I would have to read it first but I believe that a paper that follows a structure similar to what I outlined above could be quite interesting. Such a rewrite is obviously a lot of work so I'm not going to request the authors get it done within the rebuttal time period, but sincerely request they do so afterwards.

---

> > > ### Author Response · Authors · 2024-12-02
> > >
> > > >1. First, acknowledge that if we think of the linear version of the loss without the reparameterization trick / log det Jacobian correction, one can at best hope to do as well as BC. You can point out that in theory if one samples enough, they will literally get the BC gradient. You've already worked out the math required for this in the Gaussian setting above. I would suggest also implementing this on some environment to confirm this fact empirically.
> > >
> > >
> > > We would like to clarify that the mathematical derivations you refer to were **part of our discussions during the rebuttal process and are not our focus, i.e., analyzing a linear version and reframing design decisions are out of scope of this paper**. Our work inherits CPO's log sigmoid and adopts the reparameterization trick from SAC's squashed Gaussian policy to address the challenges in continuous control (lines 114-116 and 197-201). Hence, implementing and empirically testing the linear version are not our focus.
> > >
> > > ___
> > > >2. You can then point out that your "key contribution" is a series of design decisions to unlock the benefits of learner-generated negative samples that the naive approach described above is unable to do. Being honest here will only make your work seem more well thought-out / nuanced / interesting: point 1. can be spun as a "feature" not a '"bug" for your paper.
> > >
> > > Thank you for your suggestions, and we appreciate your perspective of our work. However, we believe that our current presentation effectively communicates the objectives and contributions of our study. **Including additional discussions and issues on this topic would be beyond the intended scope of our paper** (may also cause us to exceed the page limit). This could be an interesting direction for future work.
> > >
> > > ___
> > > >3. I would begin by observing that in contrast to the way people implement things for SPIN / online DPO, you keep the gradients flow to the learner generated samples. This is the first key difference that us leading to something different than the BC gradient as I observed above. I would strongly suggest independently ablating this change independently on top of the "linear" loss.
> > >
> > > **As mentioned above, our work specifically focuses on continuous control tasks with bounded action spaces.** While exploring POIL's behavior in other domains (linear version) and conducting additional ablation studies could be interesting, we consider these as future work directions, as they extend beyond our current focus.
> > > ___
> > > >4. I would then observe that drawing upon the rich literature on direct alignment algorithms, another way to differentiate yourself from BC is by adding in a nonlinearity on top of the likelihood difference (any monotonic link function should be ok in theory, one of which is $\sigma$). I would suggest applying each of these "link functions" to BC, POIL without rsample, and POIL with rsample. I would expect the first two options to perform similarly. You can then argue that it is the combination of these two changes that leads to POIL's improvement in performance.
> > >
> > > While additional analysis of different nonlinearities could be interesting, we consider this as future work as it requires substantial additional experiments beyond our current focus.

---

> > > > ### Author Response · Authors · 2024-12-02
> > > >
> > > > >4. (cont.) I don't think merely pointing out the gradients, as you did above, is sufficient here -- we'd need to see empirically how much of a performance improvement there is.
> > > >
> > > > >5. You can then pick the best link function and scale things up across multiple environments, reporting either the last or minimum validation error rather than the maximum over training, which can hide a variety of instabilities. More D4RL environments could be good to look into here. Given the details on which nonlinearity one chooses / the use of rsample matter here (i.e. there's no clear theory reason one should be better than another AFAIK), I think the burden of proof is on the authors to show empirically that their choices scales to multiple harder environments.
> > > >
> > > > **We have already conducted additional experiments on Adroit enviroments. Our empirical results already demonstrate POIL's consistent superior performance over BC across all experiments, including the additional challenging Adroit manipulation tasks in #Reply to all reviewer, also in appendix A.**
> > > >
> > > > ---
> > > > > That said, I don't think the paper currently does a good job of carefully ablating what actually differentiates POIL from BC: independently, the negative samples do nothing, one necessarily has to combine them with things like rsample and nonlinearities to see a difference (which could be good or bad).
> > > >
> > > > We have shown the differences between POIL and BC in continous control condition, which is the subject we're focusing in this paper (lines 114-116 and 197-201). Further exploration about POIL's behavior and different loss format has exceeded the scope of this paper. We will leave it as valuable future work.
> > > >
> > > > ___
> > > > > Furthermore, I do not think it is acceptable to report results where one takes the max over training in IL, so it is hard for me to give points for the current set of results in Table 2. This makes it difficult for me to fully believe the two differentiating factors from BC are generally a good idea.
> > > >
> > > > Thank you for the request. Like we replied earlier, since this requires additional experiments for a purely offline model selection procedure and checkpoints, we are not able to provide these soon, but expect to give in the next version.
> > > > ___
> > > > >I would have to read it first but I believe that a paper that follows a structure similar to what I outlined above could be quite interesting. Such a rewrite is obviously a lot of work so I'm not going to request the authors get it done within the rebuttal time period, but sincerely request they do so afterwards.
> > > >
> > > > Thank you for your suggestions regarding the structure of our paper. We appreciate your interest in our work and your ideas on how it could be further developed, while such a significant rewrite would require substantial additional work and a shift in focus.

---

### Official Review · Reviewer_ph2k · 2024-11-03

**Soundness:** 2
**Presentation:** 3
**Contribution:** 3
**Rating:** 6
**Confidence:** 3

**Summary:**

This paper introduce an imitation learning method called POIL, a DPO-like method, that compares agent actions (negative example) with expert actions (positive example). It evaluates POIL on 3 Mujoco tasks: halfcheetah, hopper, and walker2D.

**Strengths:**

- POIL is an interesting application of DPO like methods to robotics.
- the results demonstrate sample-efficiency gains compared to a few IRL and one BC baseline on Mujoco environments
- the ablations comparing various DPO like methods in robotics is interesting

**Weaknesses:**

- This work is missing comparisons to (or atleast discussions in related work comparing to) sample-efficient BC approaches that can work with one or very few demonstrations like ROT [1] and MCNN [2].
- A comparison (or discussion about) CPL [3] and other baselines in the CPL paper, another RLHF method for robotics, is also missing.
- The evaluation environments are very simple vector-observation mujoco environments. It would be helpful to extend to either more environments from D4RL that testing stitching like the ant maze environment, or more complicated image-based environments like Atari, or more dexterous robotics environments like Robosuite and Adroit.
- the return tabulated for different methods is not normalized --- this makes it hard to determine its performance between random and expert and hard to compare with other papers.


[1] S Haldar, et al, Watch and match: Supercharging imitation with regularized optimal transport, CoRL 22

[2] K Sridhar, et al, Memory-Consistent Neural Networks for Imitation Learning, ICLR 24

[3] J Hejna, et al, Contrastive Preference Learning: Learning from Human Feedback without RL, ICLR 24

**Questions:**

Please see weaknesses.
Also, have the authors investigated a provable upper bound on the sub-optimality gap for POIL, maybe building on guarantees for DPO-like methods?

---

> ### Author Response · Authors · 2024-11-19
>
> ### Reply to Weaknesses
> > This work is missing comparisons to (or atleast discussions in related work comparing to) sample-efficient BC approaches that can work with one or very few demonstrations like ROT [1] and MCNN [2].
>
>
> Thank you for your suggestions. While ROT is online learning approach makes direct empirical comparisons challenging given POIL's offline setting, we conduct new experiments with MCNN on the Adroit environment. Using the same D4RL dataset, our preliminary results show competitive performance, which has been shown on the section of "Reply to All Reviewers".
> ___
>
> > A comparison (or discussion about) CPL [3] and other baselines in the CPL paper, another RLHF method for robotics, is also missing.
>
> While both POIL and CPL address robotic control through RLHF approaches, they operate under different learning settings - CPL leverages suboptimal demonstrations and preference learning, while POIL focuses on learning from expert demonstrations. Given these different learning conditions, establishing a fair empirical comparison would be challenging.
> ___
>
> > the return tabulated for different methods is not normalized –- this makes it hard to determine its performance between random and expert and hard to compare with other papers.
>
> We understand the concern regarding normalization of returns for easier comparison. In our paper, we specifically noted that the scores are not normalized to expert data because we cannot directly obtain the expert scores from this dataset. Given this constraint, we report the raw scores to maintain transparency across all experimental conditions.
> ___
>
> > The evaluation environments are very simple vector-observation mujoco environments. It would be helpful to extend to either more environments from D4RL that testing stitching like the ant maze environment, or more complicated image-based environments like Atari, or more dexterous robotics environments like Robosuite and Adroit.
>
> We have extended our experiments to include the **Adroit** tasks from the D4RL benchmark suite, which has been shown on the section of "Reply to All Reviewers".
>
> Due to the limited time for the rebuttal, we focused on the expert datasets within these tasks, using the same settings as in the MCNN paper to ensure consistency and fair evaluation. We set $\lambda=0$ and $\beta=1$ in our experiments. Our updated results, presented in Tables 1 and 2 above, demonstrate that **POIL outperforms other state-of-the-art methods**, including those discussed in the MCNN paper, across all evaluated tasks.
>
> These results validate the scalability and effectiveness of POIL in more complex and realistic environments, addressing your concern. We appreciate your suggestion, as it has allowed us to further demonstrate the strengths of POIL. We will include these results in the Appendix of our next revised paper.

---

> > ### Comment · Reviewer_ph2k · 2024-11-20
> > **Thanks**
> >
> > Thank you. My concerns have been addressed. I have updated my score to 6.

---

> > > ### Author Response · Authors · 2024-11-22
> > >
> > > Thank you very much for your positive and encouraging feedback toward acceptance. We are pleased that our responses addressed your concerns. Your valuable comments, particularly regarding the experiments, will greatly help us improve the quality of our revision.

---

### Official Review · Reviewer_xU3D · 2024-11-04

**Soundness:** 3
**Presentation:** 2
**Contribution:** 2
**Rating:** 6
**Confidence:** 3

**Summary:**

This paper considers a practical offline imitation learning setting, aiming at learning the agent’s policy from limited demonstration without environmental interactions. The proposed method is called Preference Optimization for Imitation Learning (POIL), which compares the agent’s actions with the expert’s actions and computes the preference loss for updating the policy parameters. The empirical results on MuJoCo show consistently superior performances compared to several offline imitation learning baselines.

**Strengths:**

1.	POIL adapts the preference optimization techniques from large language models, eliminating the need for preference datasets, a discriminator, or a preference model. The learning process is simplified, avoiding adversarial training instability. This work exemplifies the successful adaptation of DPO-like alignment methods in LLMs to control problems in reinforcement learning and imitation learning.
2.	POIL is simple and effective and shows superior performance in MuJoCo tasks compared to several offline imitation learning methods. POIL also performs better than other preference optimization methods in MuJoCo tasks.

**Weaknesses:**

1.	From Section 2, we know that SPIN is the most closely related to the proposed POIL, however, the POIL objective is directly adapted from CPO instead of SPIN, which is strange and may cause confusion. I believe CPO is chosen because it avoids using a preference model. In this sense, maybe CPO is the most related work to POIL?
2.	SPIN can generate its training data and refine itself by distinguishing between current and previous outputs, continuously updating its reference model. In Section 3.2, Equation (3) is explained to maximize the divergence between the agent’s current behavior and its previous, which tries to link POIL to SPIN. However, it is not clear how this goal has been achieved.

**Questions:**

1.	How does equation (3) maximize the divergence between the agent’s current and previous behaviors?
2.	It seems POIL works quite well when \lambda=0 in Table 1, and POIL is sensitive to different \lambda in Figure 3. Does it mean that there is no need to introduce the BC regularization to the POIL loss?

---

> ### Author Response · Authors · 2024-11-19
>
> ### Reply to Weaknesses
> > From Section 2, we know that SPIN is the most closely related to the proposed POIL, however, the POIL objective is directly adapted from CPO instead of SPIN, which is strange and may cause confusion. I believe CPO is chosen because it avoids using a preference model. In this sense, maybe CPO is the most related work to POIL?
>
> Thank you for your question regarding the relationship between POIL, SPIN, and CPO. Let us clarify as follows.
>
> - **Clarifying the Relationship:**
> While SPIN and CPO both inspire our work, they contribute in different ways:
>     - **SPIN (Self-Play Fine-Tuning)** introduces a mechanism where a model improves itself by comparing its current outputs with those from a previous version (the reference model), continuously updating the reference model during training. This concept of self-improvement without external data aligns with our goal in POIL.
>     - **CPO (Contrastive Preference Optimization)** provides a reference-free loss function by eliminating the need for a reference policy, which simplifies the optimization process. This aligns with our objective to avoid the complexities associated with maintaining a reference model.
> - **Why We Adopted the CPO Objective:**
>     In POIL, we adapt the reference-free loss function from CPO because it allows us to directly compare the agent’s actions with expert actions without needing a reference model or preference dataset. This simplifies the training process and reduces computational overhead.
> - **How SPIN Influences POIL:**
>     Although we use the CPO loss function, the concept of self-improvement in POIL is inspired by SPIN. Specifically, we draw from SPIN’s idea of refining the policy by comparing its outputs, but we modify it to compare the agent’s actions with expert actions instead of previous outputs.
>
> ---
> > SPIN can generate its training data and refine itself by distinguishing between current and previous outputs, continuously updating its reference model. In Section 3.2, Equation (3) is explained to maximize the divergence between the agent’s current behavior and its previous, which tries to link POIL to SPIN. However, it is not clear how this goal has been achieved.
>
> Thank you for pointing out the problem. We will update accordingly in our next revision, and briefly summarize as follows.
>
>
> The primary goal of Equation (3) in POIL is to **maximize the preference of expert actions over the agent’s current actions**, not to maximize the divergence between the agent’s current behavior and its previous behavior.
> - **Actual Mechanism in POIL:**
>     In POIL, we directly compare the agent's actions with the expert's actions at the same state. The loss function encourages the agent to assign higher probabilities to expert actions and lower probabilities to its own sampled actions. This is achieved through the preference loss:
> $$
> \mathcal{L}\_{\text{POIL}}(\pi\_{\theta}) = -\mathbb{E}\_{(s, a\_E, a) \sim \mathcal{D}\_E} \left[ \log \sigma\left( \beta \left( \log \pi\_{\theta}(a\_E \mid s) - \log \pi\_{\theta}(a \mid s) \right) \right) \right].
> $$
>     - The term $\log \pi_{\theta}(a_E|s)$ encourages the policy to prefer expert actions.
>     - The term $\log \pi_{\theta}(a|s)$ penalizes the probability of the agent's own (potentially suboptimal) actions.
> - **Link to SPIN:**
>     While we do not maximize the divergence between the agent’s current and previous behaviors as in SPIN, we are inspired by SPIN’s self-improvement through preference comparisons. In POIL, instead of using a previous policy as a reference, we use the expert’s actions as the preferred behavior, and the agent’s own actions as the less preferred behavior. This adaptation allows us to refine the agent’s policy by learning from the discrepancies between its actions and the expert’s actions.

---

> ### Author Response · Authors · 2024-11-19
>
> ___
>
> ### Reply to Questions
> > 1. How does Equation (3) maximize the divergence between the agent’s current and previous behaviors?
>
> The loss function in Equation (3) encourages the agent to:
> - **Increase the probability of expert actions** $\log \pi_{\theta}(a_E|s)$.
> - **Decrease the probability of its own sampled actions** $\log \pi_{\theta}(a|s)$.
>
> By doing so, the agent learns to align its policy more closely with the expert's behavior while reducing the likelihood of current actions.
>
> ___
> > 2. It seems POIL works quite well when $\lambda = 0$ in Table 1 (In this paper), and POIL is sensitive to different $\lambda$ in Figure 3. Does it mean that there is no need to introduce the BC regularization to the POIL loss?
>
> As we mentioned in Discussion Section (line 508-520), the experimental results in **Table 1** (In this paper) show, in data-scarce settings (e.g., learning from a single demonstration), setting $\lambda = 0$ yields better performance. This suggests that the BC regularization term may not be necessary in such scenarios, as it can overly constrain the policy, limiting exploration when data is limited.
>
> Conversely, as our experimental results in **Table 2** (In this paper) indicate, in environments with more abundant data, the impact of $\lambda$ is less pronounced. Both $\lambda = 0$ and $\lambda = 1$ perform competitively, indicating that the BC regularization can stabilize training without significantly impacting performance in richer data scenarios.
>
> The inclusion of the BC regularization term $\lambda \cdot \mathcal{L}_{\text{BC}}$ is intended to encourage the policy to stay close to the expert actions, similar to how previous works like [1] and [2] incorporate behavior regularization in offline reinforcement learning. This helps prevent bootstrapping on out-of-distribution actions, which can be critical in data-rich environments. However, in data-limited scenarios, this term might restrict the policy too much, preventing adequate exploration and generalization.
>
> Therefore, while the BC regularization can be beneficial in certain cases, it is not always necessary. The sensitivity to $\lambda$ highlights the importance of tuning this hyperparameter based on the amount and quality of the available expert data. We will expand on this point in the revised manuscript and provide more detailed guidance on setting $\lambda$ based on different data availability contexts.
>
>
>
> [1] Fujimoto, S., & Gu, S. S. (2021). A minimalist approach to offline reinforcement learning. Advances in neural information processing systems, 34, 20132-20145.
> [2] Wu, Y., Tucker, G., & Nachum, O. (2019). Behavior regularized offline reinforcement learning. arXiv preprint arXiv:1911.11361.

---

> > ### Author Response · Authors · 2024-11-22
> >
> > Thank you for your detailed review and feedback. We hope our responses have effectively addressed your questions and concerns. If there are any remaining aspects requiring clarification, we would be happy to provide further details. We trust that the additional information has clarified the strengths of our work, and we hope this may lead to a favorable reconsideration.

---

> > > ### Comment · Reviewer_xU3D · 2024-11-26
> > > **Respond to authors' updates**
> > >
> > > Thank you for your response. Most of my concerns have been addressed. I increase my score to 6.

---

> > > > ### Author Response · Authors · 2024-11-27
> > > >
> > > > Thank you very much for your positive and encouraging feedback toward acceptance. We are pleased that our responses addressed your concerns. Your valuable comments will greatly help us improve the quality of our revision.

---

### Author Response · Authors · 2024-11-19
**Reply to All Reviewers**

Dear reviewers:

Thank you for your thoughtful reviews and for highlighting important aspects of our work. We also appreciate your recognition of the strengths of our proposed method, POIL, and we value your feedback that requires clarification. Several of you suggested extending our experiments to more complex environments (e.g. Adroit) to demonstrate the scalability and robustness of POIL.

Due to the limited time for the rebuttal, we focused on the expert dataset within the Adroit tasks mentioned in the MCNN paper to ensure consistency and fair evaluation. We used the same settings to benchmark POIL directly against MCNN under similar conditions. Specifically, we set $\lambda=0$ and $\beta=1$ in our experiments.

As shown in Tables 1 and 2 below, our method, **POIL ($\lambda=0$, $\beta=1$)**, outperforms all methods in the MCNN paper in all cases, except for the `hammer-expert-v1` task when using the full dataset (5,000 demonstrations). In the limited dataset scenarios presented in Table 2, POIL consistently achieves superior performance across all tasks.

**Table 1: Performance Comparison on Adroit Tasks**

| Task                | D4RL BC | BeT-BC      | 1-NN          | VINN          | CQL-Sparse     | Implicit BC   | MLP-BC (ReImpl.) | MCNN + MLP (Fixed Hypers) | MCNN + MLP (Tuned Hypers) | **POIL (ours)**     |
|---------------------|---------|-------------|---------------|---------------|----------------|---------------|-------------------|---------------------------|---------------------------|-----------------|
| **pen-expert-v1**       | 2633    | 1853 ± 117  | 3102 ± 225    | 3157 ± 88     | 671 ± 14       | 2586 ± 65     | 3194 ± 127       | 3285 ± 209                | 3405 ± 328                | **4077 ± 66**   |
| **hammer-expert-v1**    | 16140   | 2731 ± 261  | 13496 ± 140   | 10551 ± 1010  | 11311 ± 502    | -132 ± 25     | 13710 ± 202      | 16027 ± 382               | 16387 ± 682               | **16295 ± 49**  |
| **door-expert-v1**      | 969     | 356 ± 35    | 2761 ± 165    | 2760 ± 2.6    | 2731 ± 117     | 361 ± 67      | 2789 ± 41        | 3033 ± 0.3                | 3035 ± 7.0                | **3040 ± 12**   |
| **relocate-expert-v1**  | 4289    | 490 ± 42    | 1095 ± 268    | 1283 ± 123    | 1910 ± 1168    | -             | 4361 ± 55        | 4566 ± 47                 | 4566 ± 47                 | **4606 ± 45**   |

**Table 2: Performance with Varying Number of Demonstrations**

| Task            | Method      | 100 demos             | 500 demos             | 1000 demos            | 2000 demos            | 4000 demos            | 5000 demos (full dataset) |
|-----------------|-------------|-----------------------|-----------------------|-----------------------|-----------------------|-----------------------|---------------------------|
| **door-expert-v1**  | MCNN+MLP    | 2724.61 ± 138.82      | 2931.19 ± 31.84       | 2991.88 ± 18.31       | 3016.65 ± 9.83        | 3025.47 ± 3.31        | 3035 ± 7.0                |
|                 | **POIL (ours)** | **3015.25 ± 3.24**    | **3025.14 ± 0.85**    | **3026.52 ± 7.15**    | **3028.21 ± 22.06**   | **3032.96 ± 2.83**    | **3040.96 ± 11.99**       |
|                 |             |                       |                       |                       |                       |                       |                           |
| **pen-expert-v1**   | MCNN+MLP    | NA                    | 3711.79 ± 31.6        | 3807.97 ± 5.58        | 3857.74 ± 29.16       | 3933.75 ± 41.83       | 4051 ± 195                |
|                 | **POIL (ours)** | **4146.43 ± 104.07**  | **4126.53 ± 8.31**    | **4141.03 ± 38.12**   | **4196.78 ± 135.55**  | **4172.38 ± 113.10**  | **4077.85 ± 66.01**       |

These results demonstrate that **POIL not only scales to more complex and realistic environments but also maintains superior performance compared to existing methods**. We believe this addresses concerns about the applicability of POIL beyond the MuJoCo benchmarks and further validates the potential of our method in practical applications.

We will incorporate these new experimental results into the next revision of our paper to highlight the scalability and effectiveness of POIL in more realistic settings.

Authors

---

### Author Response · Authors · 2024-11-29
**Summary of Revisions and Response to Reviewer Feedback**

Dear Reviewers,

We would like to thank all of you for your valuable feedback and insightful comments on our manuscript. We have considered your suggestions and have made a revision to address the points raised, in which we have incorporated the changes highlighted in **red text** for your convenience. Below is a summary of the main updates:

1. **Clarification on the Use of the Reparameterization Trick:**
   - In Section 3.3 (Algorithm), we have added a detailed explanation of how we employ the *reparameterization trick* when sampling actions from the policy. Specifically, we model the policy as a Gaussian distribution with mean $\mu_{\theta}(s)$ and standard deviation $\zeta_{\theta}(s)$, and sample actions using:
     $$
     a = \tanh\left(\mu_{\theta}(s) + \zeta_{\theta}(s) \cdot \epsilon\right),
     $$
     where $\epsilon \sim \mathcal{N}(0, I)$. This ensures that gradients flow through the sampling process during optimization, addressing concerns about differentiability.
   - In Algorithm 1, we have revised the action sampling step to explicitly state accordingly that agent actions are generated via **reparameterized sampling** from the policy $\pi_{\theta}(a|s)$. This clarifies how the sampled actions depend on the policy parameters $\theta$.


3. **Additional Experiments on Adroit Tasks:**
   - We have included a new appendix (Appendix A) presenting additional experiments on the Adroit tasks from the D4RL benchmark. These experiments demonstrate the scalability and robustness of POIL in more complex and high-dimensional environments. The results show that POIL outperforms existing methods on several tasks, highlighting its effectiveness beyond the MuJoCo benchmarks.

4. **Detailed Results with Error Bars:**
   - In Appendix B, we provide detailed results for the single demonstration learning experiments, including standard deviations over multiple runs. This offers a comprehensive view of the performance variability and reinforces the robustness of POIL compared to baseline methods.


We believe that these revisions address the concerns and suggestions raised by the reviewers. We are grateful for your time and constructive feedback, which have helped us improve the quality of our work.

We hope the revision meets your expectations.

Sincerely,
The Authors

---

### Meta-Review · Area_Chair_4kkv · 2024-12-21

**Metareview:**

The paper introduces Preference Optimization for Imitation Learning (POIL), an offline imitation learning algorithm inspired by preference optimization. POIL learns a policy by comparing agent and expert actions, using a preference loss without requiring preference datasets, discriminators, or adversarial training. Experimental results demonstrate strong performance, particularly in low-data settings, outperforming state-of-the-art offline imitation learning methods on various MuJoCo tasks.

Reasons to accept
- This work adapts DPO-like techniques to offline imitation learning, requiring no preference models or discriminators, offering a potentially simpler and more stable approach compared to adversarial methods.
- POIL shows improved performance compared to baselines, e.g., BC, IQ-Learn, DMIL, and O-DICE, especially in low-data regimes.

Reasons to reject
- Evaluation is restricted to simple MuJoCo environments. It is unclear if the method applies to image-based tasks or dexterous robotic tasks.
- The experimental results show the best over the training run instead of reporting average task performance, which is not standard in IL evaluations.
- After the reviewer-author discussion phase, the reviewers were still not convinced that the proposed method is not provably equivalent to a noisy version of BC.

While this paper explores a promising research direction and introduces an interesting approach, it would greatly benefit from substantial revisions to address the concerns raised by the reviewers. I suggest refining the work further and resubmitting it to another venue. Therefore, I recommend rejecting the paper in its current form.

**Additional Comments On Reviewer Discussion:**

During the rebuttal period, all five reviewers acknowledged the author's rebuttal, and two reviewers adjusted the score accordingly.

---

### Decision · Program_Chairs · 2025-01-22

Reject